# Is Factuality Enhancement a Free Lunch For LLMs? Better Factuality Can Lead to Worse Context-Faithfulness

**Baolong Bi**[1,2]   **Shenghua Liu**[1,2*]   **Yiwei Wang**[3]   **Lingrui Mei**[1,2]
**Junfeng Fang**[4]   **Hongcheng Gao**[2]   **Shiyu Ni**[2]   **Xueqi Cheng**[1,2]
[1]CAS Key Laboratory of AI Safety, Institute of Computing Technology, CAS
[2]University of Chinese Academy of Sciences [3]University of California, Merced
[4]University of Science and Technology of China (USTC), Hefei, China
{bibaolong23z, liushenghua, nishiyu23z, cxq}@ict.ac.cn, wangyw.evan@gmail.com
{meilingrui22, gaohongcheng23}@mails.ucas.ac.cn, fjf@mail.ustc.edu.cn

## Abstract

As the modern tools of choice for text understanding and generation, large language models (LLMs) are expected to accurately output answers by leveraging the input context. This requires LLMs to possess both context-faithfulness and factual accuracy. While extensive efforts aim to reduce hallucinations through factuality enhancement methods, they also pose risks of hindering context-faithfulness, as factuality enhancement can lead LLMs to become overly confident in their parametric knowledge, causing them to overlook the relevant input context. In this work, we argue that current factuality enhancement methods can significantly undermine the context-faithfulness of LLMs. We first revisit the current factuality enhancement methods and evaluate their effectiveness in enhancing factual accuracy. Next, we evaluate their performance on knowledge editing tasks to assess the potential impact on context-faithfulness. The experimental results reveal that while these methods may yield inconsistent improvements in factual accuracy, they also cause a more severe decline in context-faithfulness, with the largest decrease reaching a striking 69.7%. To explain these declines, we analyze the hidden states and logit distributions for the tokens representing new knowledge and parametric knowledge respectively, highlighting the limitations of current approaches. Our finding highlights the complex trade-offs inherent in enhancing LLMs. Therefore, we recommend that more research on LLMs' factuality enhancement make efforts to reduce the sacrifice of context-faithfulness.

## 1 Introduction

Leveraging their powerful capabilities in text understanding and generation, large language models (LLMs) can effectively integrate contextual information to infer and generate truthful responses (OpenAI, 2023; Touvron et al., 2023a;b; Song et al., 2024). However, hallucinations (Huang et al., 2023; Tonmoy et al., 2024) significantly undermine the LLMs' reliability, primarily due to inadequate adherence to contextual instructions and failure to produce truthful responses (Zhang et al., 2023b).

With the widespread adoption of Retrieval-Augmented Generation (RAG) (Fan et al., 2024; Santhanam et al., 2021; Schick et al., 2024; Qin et al., 2024), the study of context-faithfulness in LLMs is becoming increasingly important (Chen et al., 2022; Li et al., 2024c; Bi et al., 2024a). This is because they depend on the effective integration of retrieved external information into their generated responses to ensure contextual relevance. Therefore, exceptional LLMs must not only produce factual outputs based on their parametric knowledge but also comply with contextual requirements. Context-faithfulness enhances LLMs' compliance with user instructions and improves performance, particularly when the parametric knowledge is insufficient or outdated.

---

*  Corresponding author.

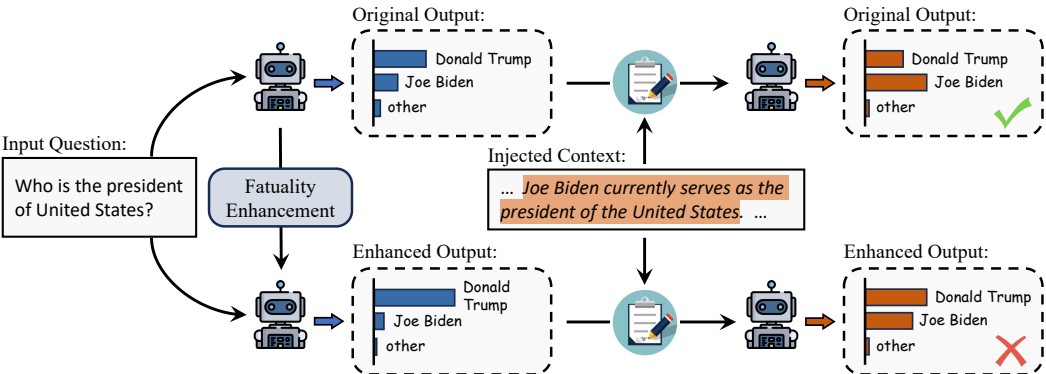

Figure 1: Comparison of responses from native LLMs and factuality-enhanced LLMs before and after context injection. The enhanced LLMs, due to overconfidence in their parametric knowledge, struggle to integrate new information from the context, resulting in incorrect answers.

Prior work (Li et al., 2024a; Zhang et al., 2024c) suggests that the cause of factual hallucinations is LLMs' inability to accurately "convey" knowledge during inference, even when it has been effectively "learned" during pretraining. Some emerging works focuses on narrowing the gap between "knowing" and "telling" in LLMs to improve LLMs' factuality. These approaches can be divided into two categories: one involves editing internal representations, such as attention mechanisms (Li et al., 2024b; Chen et al., 2024) or hidden states (Zhang et al., 2024b), and the other modifies the logits of tokens during decoding (Li et al., 2023b; Chuang et al., 2023; Zhang et al., 2023a). These methods enable LLMs to output their knowledge more accurately, reducing hallucinations, without altering parameters, as they do not require introducing new factual knowledge through supervised fine-tuning (SFT) (Ovadia et al., 2023) or RLHF (Bai et al., 2022).

However, is this factuality enhancement a free lunch for LLMs? We argue that while factuality enhancement can reduce factual hallucinations in LLM outputs, it may lead to excessive confidence in their parametric knowledge, making it difficult for them to comply with contextual information, especially when external knowledge contradicts their parametric knowledge (Petroni et al., 2020; Si et al., 2023; Xie et al., 2024). Figure 1 illustrates a simple example of the scenario we envisioned. In this work, we first revisit several popular methods for factuality enhancement and evaluate their performance on TruthfulQA (Lin et al., 2021). The results show that while most methods provide some improvement in factuality, some approaches exhibit minimal or inconsistent gains. Therefore, we suspect that current factuality enhancement methods, despite improving factuality, might negatively impact context-faithfulness to a greater extent.

To test our hypothesis, we evaluate these factuality enhancement methods on knowledge editing (KE) (Sinitsin et al., 2020; De Cao et al., 2021; Mitchell et al., 2022; Yao et al., 2023b) benchmarks, comparing them against the baseline performance of LLAMA2-CHAT. Specifically, we conduct experiments with in-context editing (ICE) (Zhong et al., 2023; Zheng et al., 2023; Bi et al., 2024c) methods on the MQUAKE datasets. The accuracy of ICE effectively reflects the context-faithfulness of LLMs, as it provides outdated or counterfactual contextual knowledge that may contradict the models' parametric knowledge. Experimental results indicate that, compared to the baseline, current popular factuality enhancement methods lead to a significant decline in LLM performance in context-faithfulness. For example, the editing accuracy of TruthX (Zhang et al., 2024b) on LLAMA2-7B-CHAT decreased by as much as 60.3%. This suggests that while factuality enhancement aids LLMs in accurately conveying factual information, it also introduces the risk of overconfidence, resulting in a substantial setback for the context-faithfulness of LLMs.

We explain the decline in context-faithfulness by analyzing the hidden states and token-wise distributions of LLMs' outputs, based on knowledge tokens captured by our specially designed algorithm. Compared to the baseline, the hidden states become more concentrated and closely align with the original parametric outputs after inserting contextual information. Additionally, the logits for tokens representing new knowledge decrease, while the logits for parametric knowledge increase.

Altogether, our study highlights the significant risks posed by current factuality enhancement methods, as we validate and explain their notable decline in context-faithfulness. This demonstrates the

importance of ensuring that LLMs not only enhance factual accuracy but also maintain a strong alignment with the provided context to minimize the occurrence of hallucinations. We strongly advocate for the development of LLMs that prioritize both faithful adherence to context and the accurate conveyance of factual information. Therefore, we recommend future research explore strategies to balance these two critical aspects.

## 2 PRELIMINARY

This section revisits the core concepts of language modeling (Section 2.1) and knowledge editing (Section 2.2) of LLMs, with the aim of conducting a more in-depth exploration of existing factuality enhancement strategies (Section 3) and their interpretability (Section 5), as well as the use of knowledge editing to assess context-faithfulness (Section 4).

### 2.1 LANGUAGE MODELING

The current language modeling of LLMs aims to generate text based on the probability distribution of token sequences. Formally, given a sequence of tokens $x_t = \{x_1, x_2, \ldots, x_{t-1}\}$, the goal is to predict the next token $x_t$ by modeling the conditional probability $\mathbb{P}(x_t|x_{<t})$.

To achieve this, LLMs process the input token sequence through layers of attention (Attn) and feed-forward networks (FFN). The hidden state is computed as follows:

$$\mathbb{H}_t = \text{FFN}(\text{Attn}(\mathbf{X}_{<t})) \tag{1}$$

Here, $\mathbf{X}_{<t}$ represents the embedded representation of the input tokens $x_{<t}$, which is passed through Attn to capture contextual relationships, followed by a FFN to produce the hidden state.

Then, the model can predict the probability distribution of the next token $x_t$ over the vocabulary set $\mathcal{V}$ by applying a linear transformation $\phi(\cdot)$ followed by a softmax function:

$$\mathbb{P}(x_t|x_{<t}) = \text{softmax}(\phi(\mathbb{H}_t)), \quad x_t \in \mathcal{V} \tag{2}$$

The softmax function converts the output logits from $\phi(\mathbb{H}_t)$ into a probability distribution over the vocabulary $\mathcal{V}$, allowing the model to assign a likelihood to each possible next token. At decoding time, various decoding strategies can be applied at each step $i$ to select the next token $x_i$, with the given predicted distribution of the next token $\mathbb{P}(x_t|x_{<t})$. This iterative process continues until the sequence generation reaches a designated end token or meets a predefined stopping condition.

### 2.2 KNOWLEDGE EDITING

Context-faithfulness requires LLMs to effectively adhere to external context. This means they must fully accept new knowledge provided by that context, even if it conflicts with their parametric knowledge acquired during pretraining, which may be insufficient or outdated. Therefore, this paper uses the success rate of KE to assess the context-faithfulness of LLMs, as KE provides counterfactual new knowledge that contradicts the LLMs' own parametric knowledge.

KE aims to efficiently adjust the behavior of the original model $f_{base}$ into post-edit model $f_e$ with a specific editing descriptors $z_e$. The editing descriptors $z_e$ describe a desired change in model behavior and can be represented as $z_e = (x_e, r_e, y_e)$, where $(x_e, r_e, y_e)$ represents a triplet such as (*US*, *President*, *Joe Biden*) meaning Joe Biden is the president of US. The ultimate objective of KE is to edit the LLM output that $f_e(x_e, r_e) = y_e$ while $f_{base}(x_e, r_e) \neq y_e$.

A thorough edit should not only modify the relevant knowledge but also update all knowledge affected by this edit in multi-hop relationships. For instance, consider the two-hop fact triple (*WWE Velocity*, *created by*, *Vince McMahon*) and (*Vince McMahon*, *spouse*, *Linda McMahon*). With a fact edit $z_e = (WWE\ Velocity, created\ by, Stan\ Lee)$ and an additional fact (*Stan Lee*, *spouse*, *Joan Lee*), the correctly updated answer should be *Joan Lee*. We adopt a multi-hop editing task to measure context-faithfulness of LLMs in Section 4 by evaluating the accuracy of multi-hop question answering with fact edits. This poses challenges for LLMs in adhering to the editing context due to potential conflicts between new knowledge and their parametric knowledge during the reasoning process.

| Model | Method | TruthfulQA | | | FACTSCORE | | |
|---|---|---|---|---|---|---|---|
| | | MC1 (%) | MC2 (%) | MC3 (%) | Resp. (%) | Facts (#) | Score (%) |
| | Baseline | 37.6 | 54.6 | 28.1 | 37.5 | 45.7 | 63.8 |
| LLaMA2-7B-CHAT | DoLa♠ | **32.9** | 60.8 | 29.5 | 40.7 | 48.7 | **61.3** |
| | ICD♠ | 46.3 | 69.1 | 41.2 | **36.1** | 46.6 | 66.3 |
| | ITI◇ | **37.0** | 54.7 | **27.8** | 41.9 | **40.8** | **62.4** |
| | TruthX◇ | 54.2 | 73.9 | 44.3 | 40.6 | **43.7** | 65.3 |
| | Baseline | 37.7 | 55.6 | 28.2 | 77.0 | 37.6 | 52.5 |
| LLaMA2-13B-CHAT | DoLa♠ | **37.3** | 61.8 | 32.7 | **55.8** | **46.7** | 59.8 |
| | CD♠ | **28.2** | **54.9** | 29.8 | **74.2** | 39.8 | 53.5 |
| | ICD♠ | 45.6 | 67.5 | 42.3 | **46.9** | 43.4 | 59.5 |
| | ITI◇ | 38.9 | **53.4** | 29.6 | 78.9 | **34.5** | 55.3 |

Table 1: Experimental results of factuality evaluation on TruthfulQA and FACTSCORE. We conduct experiments with various factuality enhancement methods, including both contrastive decoding and representation editing, on LLaMA2-CHAT. Methods marked with ♠ belong to contrastive decoding, while those marked with ◇ belong to representation editing. In FACTSCORE, % Resp. indicates the response ratio of LLMs, while # Facts represents the number of extracted atomic facts per response. Here, **red** highlights a decrease in factuality compared to the Baseline.

## 3 FACTUALITY ENHANCEMENT FOR LLMS

### 3.1 CURRENT APPROACHES TO FACTUALITY ENHANCEMENT

SFT or RLHF methods(Yang et al., 2023a; Ovadia et al., 2023; Ouyang et al., 2022; Bai et al., 2022) enhance the factuality of LLMs by injecting external training data. However, this approach significantly increases costs and introduces unreliability, potentially jeopardizing the integrity of the model's inherent knowledge. Current methods for enhancing factuality do not modify the underlying structure or parameters of LLMs; instead, they focus on improving how LLMs convey the information they have already learned to reduce hallucinations. These approaches primarily involve modifying the outputs of LLMs, including the hidden states obtained through Attn and FFNs (eq. (1)), as well as the logits distribution used to predict the next token during the decoding phase (eq. (2)). The current factuality enhancement methods in generation can be classified into the following two categories based on their different approaches to modifying LLM outputs:

**Representation Editing** Representation editing methods (Burns et al., 2022; Li et al., 2024b; Zhang et al., 2024b; Chen et al., 2024) enhance factuality by editing the internal representations of LLM outputs. They learn a direction within attention heads or a potential truthful space to modify the attention patterns of the LLM, thereby adjusting the hidden states.

**Contrastive Decoding** Contrastive decoding methods (Li et al., 2023b; Chuang et al., 2023; Zhang et al., 2023a; Kai et al., 2024; Bi et al., 2024b) enhance the factuality of LLMs by comparing the output probabilities of expert and amateur models, different layers within the transformer, various tokens, and the outputs of normal and hallucination-injected models.

### 3.2 EVALUATION OF FACTUALITY

We evaluate the performance of popular factuality enhancement methods on the LLaMA2-7B-CHAT and LLaMA2-13B-CHAT models in terms of factuality of LLMs using the TruthfulQA (Lin et al., 2021) and FACTSCORE (Min et al., 2023) benchmarks. Evaluation on both benchmarks adheres to the settings of previous studies (Chuang et al., 2023; Li et al., 2024b; Zhang et al., 2023a). Using TruthfulQA, we employ a multiple-choice task where the LLM selects an answer from various correct and incorrect options, evaluated through multiple-choice accuracy (MC1, MC2, and

| Model | Method | 3-shot | 5-shot | 10-shot | COT (10-shot) |
|-------|--------|--------|--------|---------|---------------|
| LLaMA2-7B-chat | Baseline (-) | 68.5 (-) | 78.7 (-) | 81.3 (-) | 82.5 (-) |
| | DoLa♠ | 52.3 ($\downarrow$ 22.4) | 63.7 ($\downarrow$ 15.0) | 66.5 ($\downarrow$ 14.8) | 62.5 ($\downarrow$ 20.0) |
| | ICD♠ (Zhang et al., 2023a) | 50.1 ($\downarrow$ 26.9) | 61.7 ($\downarrow$ 17.0) | 64.9 ($\downarrow$ 16.4) | 60.5 ($\downarrow$ 22.0) |
| | ITI◇ (Li et al., 2024b) | 51.9 ($\downarrow$ 21.8) | 62.2 ($\downarrow$ 16.5) | 67.5 ($\downarrow$ 13.8) | 65.7 ($\downarrow$ 16.8) |
| | TruthX◇ (Zhang et al., 2024b) | 26.7 ($\downarrow$ 61.1) | 32.5 ($\downarrow$ 58.8) | 32.7 ($\downarrow$ 59.9) | 26.5 ($\downarrow$ 67.9) |
| LLaMA2-13B-chat | Baseline (-) | 78.3 (-) | 84.5 (-) | 86.1 (-) | 86.3 (-) |
| | DoLa♠ (Chuang et al., 2023) | 58.3 ($\downarrow$ 20.0) | 67.7 ($\downarrow$ 16.8) | 70.2 ($\downarrow$ 15.9) | 70.3 ($\downarrow$ 16.0) |
| | ICD♠ (Zhang et al., 2023a) | 59.5 ($\downarrow$ 23.2) | 66.2 ($\downarrow$ 18.3) | 72.4 ($\downarrow$ 13.7) | 67.6 ($\downarrow$ 18.7) |
| | ITI◇ (Li et al., 2024b) | 64.3 ($\downarrow$ 14.0) | 69.8 ($\downarrow$ 14.7) | 74.2 ($\downarrow$ 11.9) | 65.7 ($\downarrow$ 20.6) |
| | CD♠ (Li et al., 2023b) | 55.2 ($\downarrow$ 29.5) | 62.6 ($\downarrow$ 21.9) | 67.1 ($\downarrow$ 19.0) | 64.9 ($\downarrow$ 21.4) |

Table 2: Experimental results of context-faithfulness evaluation on MQUAKE dataset. $k$-shot represents the number of context demonstrations provided, which is detailed in Figure 2 along with the use of COT. In the subscript ($\downarrow \Delta$), $\Delta$ (%) represents the decrease compared to the baseline.

MC3). For FACTSCORE, the assessment is conducted through retrieve+chatGPT methodology. The implementation details of the factuality enhancement methods are in Appendix A.

The experimental results of LLM factuality on LLaMA2-CHAT are presented in Table 1. A notable observation is that, compared to the baseline, factuality enhancement methods show improvements across most metrics. Without altering the structure or parameters of LLMs, these methods reduce the gap between 'know' and 'tell' knowledge to mitigate hallucinations, making such improvements highly valuable. However, these methods still have significant limitations. The improvement in factuality is modest and quite unstable. The red-marked sections in Table 1 indicate that the evaluated methods do not show consistent enhancements across all metrics; in fact, some metrics exhibit declines compared to the baseline. This suggest that these methods for modifying LLM outputs may negatively impact the model's natural generation capabilities.

---

[ *k contextual demonstrations abbreviated* ]

**Question:** What is the official language of the country where the creator of WWE Velocity's spouse is a citizen?

**Edit:** The creator of WWE Velocity is Stan Lee. The official language of US is German.

**Thoughts** *(if COT):* WWE Velocity was created by Stan Lee, Joan Lee is the spouse of StanLee, Joan Lee is a citizen of US, the official language of US is German.

**Answer:** German.

---

Figure 2: An illustration of the ICE task to evaluate LLMs' context-faithfulness. It involves multi-hop question answering with corresponding edits, utilizing $k$ contextual demonstrations to guide editing and align output format. The highlighted text represents the expected output from the LLMs, while *Thoughts* indicates the additional step taken when using Chain of Thought (COT) reasoning.

## 4 EVALUATION OF CONTEXT-FAITHFULNESS

The factuality enhancement methods increase LLMs' confidence in their parametric knowledge to produce more accurate outputs. However, we argue that this can lead to overconfidence, which reduces their faithfulness to context and consequently introduces new risks of hallucination. To validate this suspicion, we evaluates the impact of these factuality enhancement methods on context-faithfulness.

### 4.1 EXPERIMENTAL SETUP

**Task** We meticulously design the In-Context Editing (ICE) (Zheng et al., 2023; Zhong et al., 2023; Wang et al., 2024; Bi et al., 2024d;c) task to evaluate the context-faithfulness of LLMs. As the most effective KE method currently, ICE prompts LLMs to reason and generate answers that faithfully

incorporate new knowledge retrieved from edit memory in the form of context. We configure the edit memory to include only question-relevant editing instances, ensuring that all provided editing context influences the LLMs' outputs. We assess context-faithfulness based on the accuracy of question-answering with edits, as this reflects whether LLMs remain faithful to external context when it conflicts with their internal knowledge. Figure 2 provides a simple illustration of our task, with additional contextual demonstrations listed in the Appendix D.2.

**Models and Datasets**    We use LLaMA2-7B-CHAT and LLaMA2-13B-CHAT as baseline models to conduct experiments on MQUAKE (Zhong et al., 2023) [1], comparing various factuality enhancement methods against the natural LLaMA2-CHAT models. MQUAKE provides multi-hop knowledge questions to evaluate knowledge editing on counterfactual edits, as introduced in Section 2.2. This verifies whether knowledge has been properly edited and effectively demonstrates LLMs' context-faithfulness. CD compares the decoding logits of the 13B version of LLaMA2-CHAT with those of the 7B version, using LLaMA2-13B-CHAT as the baseline for comparison. Implementation details of the factuality enhancement methods can be found in the Appendix A.

| Model | Baseline | DoLa♠ | CD♠ | ICD♠ | ITI◇ | TruthX◇ |
|---|---|---|---|---|---|---|
| LLaMA2-7B-CHAT | 54.5 (-) | 39.4 (↓ 27.7) | - | 29.8 (↓ 45.3) | 31.5 (↓ 42.2) | 22.6 (↓ 58.5) |
| LLaMA2-13B-CHAT | 63.8 (-) | 35.6 (↓ 44.2) | 38.6 (↓ 39.5) | 39.5 (↓ 52.2) | 34.5 (↓ 45.9) | - |

Table 3: Experimental results of context-faithfulnes evaluation using MeLLo method.

## 4.2 MAIN RESULTS

Based on the above experimental setup, we evaluate the context-faithfulness of current factuality enhancement methods. The experimental results of ICE on LLaMA2-7B-CHAT and LLaMA2-13B-CHAT are shown in Table 2. As observed, all factuality enhancement methods exhibit a marked decline compared to the baseline in performance on the ICE task under any setting. Notably, this decline is not minor, with an average decrease exceeding 20%, and occurs without exception across all methods. Particu-

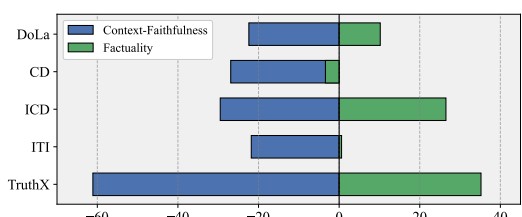

Figure 3: Changes (%) in Context-Faithfulness and Factuality of Factuality-Enhanced LLMs.

larly in the state-of-the-art TruthX for factuality, the decline reaches as high as 67.9%. This suggests that the potential risks to context-faithfulness could be significant.

An interesting observation is that the accuracy of all methods, including the baseline, improves as the number of context demonstrations increases. This aligns with the findings of Zheng et al. (2023)., suggesting that more faithful examples can enhance the context-faithfulness of LLMs. Additionally, we find that introducing Chain of Thought (COT) (Wei et al., 2022) reasoning into the baseline improves its accuracy. However, the performance of COT in factuality enhancement methods is unstable, with most methods showing a decline compared to those without COT. This may be due to the fact that during the additional COT reasoning process, LLMs with factuality enhancements tend to recall parametric knowledge more, thereby negating the new knowledge from the context. This can be observed in the cases discussed in Section 5.4.

Figure 3 visualizes the changes in both factuality and context-faithfulness of these methods compared to the baseline, based on the results of *MC2* in Table 1 and *3-shot* in Table 2. It illustrates that while these factuality enhancement methods provide unstable improvements in factuality, they also lead to a more significant decline in the context-faithfulness of LLMs.

In addition to the aforementioned basic ICE setup, we also employ the advanced KE method MeLLo (Zhong et al., 2023) to assess LLMs' context-faithfulness. MeLLo can decompose complex multi-hop questions into sequential sub-questions, checking for conflicts with relevant new knowledge

---

[1] We specifically use the released MQUAKE-CF-3K-V2 data version for our experiments, which addresses the internal knowledge conflicts within the dataset.

retrieved from edit memory to determine if edits are necessary. Therefore, we further validate context-faithfulness using MeLLo, as it performs one-hop edits, thereby avoiding hallucinations introduced by multi-hop reasoning. The prompt template and implementation details of MeLLo are provided in the Appendix D.1. The results in Table 3 show that, when using MeLLo, which is more suited for realistic editing scenarios, the accuracy of factuality-enhanced LLMs exhibits a more significant decline compared to the previous simple ICE evaluation. Most declines exceed 40% relative to the baseline, with larger models experiencing even more pronounced decreases. All of the editing results that these factuality enhancement methods significantly reduce the accuracy of knowledge editing tasks, highlighting that the current approaches severely impact the context-faithfulness of LLMs.

## 5 IN-DEPTH EXPLORATION OF CONTEXT-FAITHFULNESS

The evaluation of context-faithfulness in Section 4 confirms the earlier suspicion that improved factuality leads to a significant decline in context-faithfulness. This section aims to further explain this decline from a model interpretability perspective. We investigate the logit distributions (Section 5.2) and hidden state representations (Section 5.3) of relevant knowledge in LLM outputs captured by our algrithm (Section 5.1), and conduct case study (Section 5.4) based on the generated text, to reveal how these factuality enhancement methods impact context-faithfulness.

---

**Algorithm 1** Knowledge Token Capturing

**Require:** The LLM generates a token sequence of length $n$, $\mathcal{V}$: vocabulary of LLM, $\mathcal{P}_i$ in ($\mathcal{P}_1$, $\mathcal{P}_2$, ..., $\mathcal{P}_n$): logits distribution of tokens, $H_i$ in ($H_1$, $H_2$, ..., $H_n$): hidden states of tokens, $S_{\text{new}}$: string of new knowledge, $S_{\text{para}}$: string of parametric knowledge.
**Ensure:** Captured knowledge logits and hidden states $P_{\text{new}}$, $P_{\text{para}}$, $H_{\text{new}}$, $H_{\text{para}}$
 1: Initialize $P_{\text{new}}$, $P_{\text{para}} \leftarrow$ *None*
 2: $S_{\text{com}} = \text{COM}(S_{\text{new}}, S_{\text{para}})$         ▷ Identify common substrings
 3: **for** $\mathcal{P}_i$ in ($\mathcal{P}_1$, $\mathcal{P}_2$, ..., $\mathcal{P}_n$) **do**
 4:     **for** token $x_j$ in $\mathcal{V}$ **do**        ▷ Sort by $P_i$ in descending order
 5:         $x_j \rightarrow x'_j$         ▷ Decode $x_j$ to string $x'_j$
 6:         **if** $x'_j$ in $S_{\text{com}}$ and $P_{\text{new}} = P_{\text{para}} = None$: **break**   ▷ $x'_j$ is indistinguishable
 7:         **if** $x'_j$ in $S_{\text{new}}$ and $P_{\text{new}} = None$: $P_{\text{new}} \leftarrow P_{i,j}$, $H_{\text{new}} \leftarrow H_i$   ▷ Capture new knowledge
 8:         **if** $x'_j$ in $S_{\text{para}}$ and $P_{\text{para}} = None$: $P_{\text{para}} \leftarrow P_{i,j}$, $H_{\text{para}} \leftarrow H_i$   ▷ Capture para knowledge
 9:     **end for**
10: **end for**
11: **return** $P_{\text{new}}$, $P_{\text{para}}$, $H_{\text{new}}$, $H_{\text{para}}$

---

### 5.1 DISTINGUISHING NEW AND PARAMETRIC KNOWLEDGE IN LLM OUTPUTS

Our objective is to identify the parts of LLM outputs that distinguish between the new knowledge acquired from context and the parametric knowledge embedded within the LLMs, rather than analyzing repetitive or meaningless outputs. For example, consider an expected LLM output with injected context, such as *"Answer: United States"*, compared to the original parametric output without the injected context, which would be *"Answer: United Kingdom"*. In this case, we should not capture *"Answer:"* as it holds no factual meaning, nor should we focus on *"United"* since it is repetitive and does not reflect the difference. Instead, our focus should be on capturing tokens with clear factual significance—those that can differentiate between the new knowledge introduced by context and the parametric knowledge inherent to the model. In this example, tokens like *"Kingdom"* serve as the key markers of distinction, effectively highlighting the critical divergence between the contextual information and the model's existing knowledge.

To achieve this, we design a novel knowledge token capturing algorithm, with details presented in Algorithm 1. The algorithm captures the highest-probability tokens that differentiate new and parametric knowledge by matching the decoded tokens with the respective knowledge strings. We conduct experiments on the MQUAKE-STUBBORN dataset provided by Bi et al. (2024b), which effectively highlights the distinction between new knowledge and parametric knowledge. Specifically, we collect 2,000 question-answer instances, each consisting of the output from LLMs prior to injecting context containing new knowledge, as well as the output after the injection.

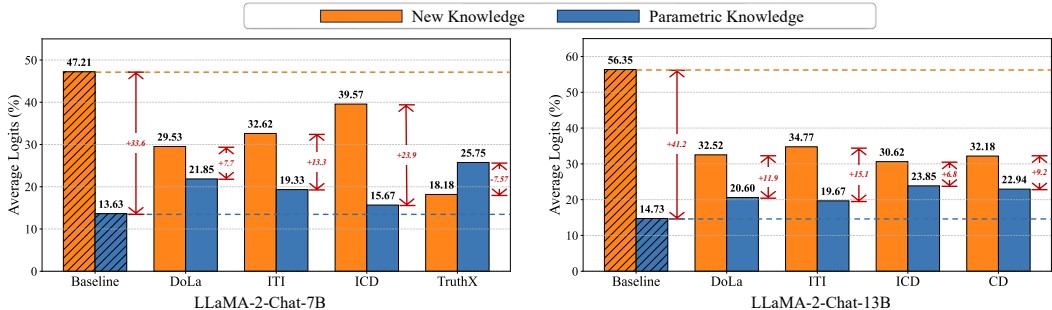

Figure 4: Average logits of tokens representing new and parametric knowledge after context injection.

## 5.2 ANALYSIS IN LOGITS DISTRIBUTION

To further investigate LLM context-faithfulness from the perspective of logits distribution (eq. (2)), we capture the tokens corresponding to both the in-context new knowledge and the parametric knowledge after context injection. The average probabilities of these tokens for the factuality enhancement methods and baselines are shown in Figure 4. We observe that, compared to the baseline, all factuality enhancement methods show a decrease in the probability of new knowledge tokens, while the probability of parametric knowledge tokens increases. This provides a clear explanation for the reduction in context-faithfulness discussed in Section 4. These methods effectively boost the retention of parametric knowledge but at the cost of diminished faithfulness to external context. Furthermore, the difference in probabilities between new and parametric knowledge correlates well with the editing accuracy in Table 2, where larger differences lead to better editing performance, while TruthX's poor editing is explained by the dominance of parametric knowledge over new knowledge.

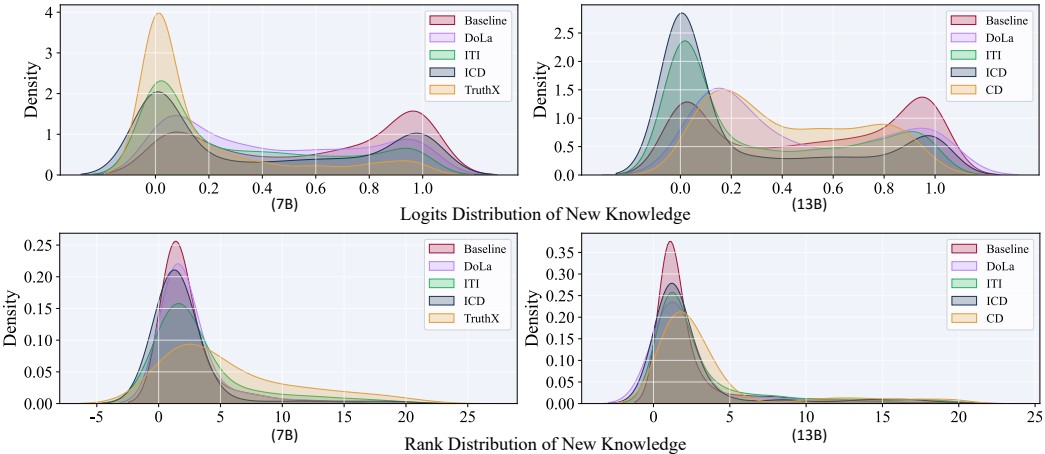

Figure 5: Statistics of new knowledge distribution for both logits and rank on LLAMA2-CHAT (left) and LLAMA2-13B-CHAT (right). Rank is recorded by capturing the position of the new knowledge token within the vocabulary $\mathcal{V}$, based on its logits ranking.

Results in figure 5 provides a deeper statistical analysis of the logits associated with new knowledge. The baselines of the LLAMA2-CHAT models consistently exhibit a higher concentration in the probability range of $0.8$ to $1.0$, with top token ranks (1 to 5) clearly dominating. In contrast, factuality enhancement methods shift the distribution of new knowledge tokens towards lower probability ranges, indicating a diminished confidence in in-context knowledge compared to the baselines.

## 5.3 ANALYSIS IN HIDDEN STATES

Section 3 shows that all factuality enhancement methods impact the logits distribution of the output, resulting in a decline in context-faithfulness. Some methods also aim to improve factuality by editing the internal representations (eq. (1)) of LLMs, which requires an analysis based on hidden states. We

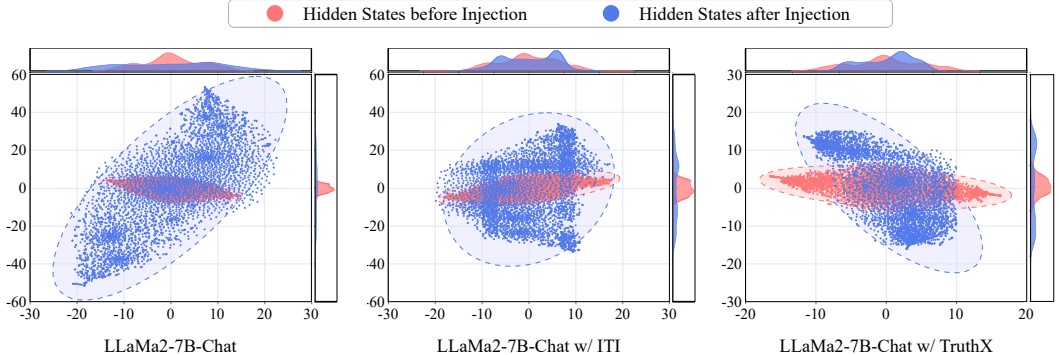

Figure 6: Distribution of reduced hidden states for new knowledge before and after context injection. The dashed circles represent the 95% confidence intervals, showing that factuality enhancement methods lead to a post-injection convergence towards the pre-injection distribution.

similarly use theknowledge token capturing algorithm to collect these crucial hidden states. Since the contrastive decoding method does not alter internal representations, our evaluation focuses on the ITI (Li et al., 2024b) and TruthX (Zhang et al., 2024b) methods. We used t-SNE to reduce the dimensionality of the hidden states representing new knowledge in LLAMA2-7B-CHAT from the original 4096 dimensions to 2 dimensions, with the perplexity set to 15.

Figure 6 presents the distribution of hidden states before and after context injection for these representation editing methods. We find that the distribution of the reduced hidden states after context injection is broader and more dispersed compared to before the injection. This indicates that context injection alters the distribution of knowledge representations, leading them to deviate from their original factuality, where the hidden states prior to injection represent the parametric factuality inherent in the LLMs. Notably, compared to the significant difference observed between the pre- and post-context injection hidden states in the baseline, the differences for ITI and TruthX are much smaller. The hidden states of new knowledge after injection are more confined and closer to those before the injection. Especially, TruthX exhibits a distribution that closely resembles the original range, which explains its significant decline in context-faithfulness. This is because it modifies both the FFN and Attn representations, making them more resistant to incorporating new knowledge and more firmly tied to the model's parametric knowledge. Overall, these representation editing methods tend to limit the natural dispersion of hidden states after context injection. As a result, the output decoded from the constrained hidden states remains more aligned with the original parametric knowledge, rather than incorporating the new contextual knowledge effectively.

## 5.4 CASE STUDY

We conduct a case study on the outputs of factuality enhancement methods in the ICE task (Section 4). We classify the context-unfaithful outputs into three categories: ignoring context, denying context, and misformatting context, with illustrative examples for each type provided in Table 4. Beyond knowledge conflicts causing the context to be overlooked or contradicted, failing to properly follow the context demonstration format also indicates a lack of context-faithfulness. Additional examples of real context-unfaithful outputs are provided in Appendix E for further examination.

## 6 RELATED WORK

**Hallucinations in LLMs**  Hallucinations (Kaddour et al., 2023; Tonmoy et al., 2024; Wang et al., 2023; Mei et al., 2024b;c) have garnered significant attention due to their pronounced side effects, leading to the generation of unreliable content (Gunjal et al., 2024; Huang et al., 2023; Liu et al., 2024; Zhang et al., 2024a). These hallucinations can arise from various sources at different stages of the LLM lifecycle. The primary reasons include conflicts between the context provided and the LLMs' inherent memory, as well as the models' inability to accurately generate knowledge that they have already acquired during pretraining (Zhang et al., 2023b; Chen et al., 2023). This paper primarily explores the interplay between these two aspects.

| Question: What is the official language of the country where the creator of WWE Velocity's spouse is a citizen? |
| --- |
| Injected Context: [k context demonstrations] Edit: The creator of WWE Velocity is Stan Lee. The official language of US is German. |

| Context-faithful output | Ignoring context | Denying context | Misformatting context |
| --- | --- | --- | --- |
| **Thoughts:** WWE Velocity was created by Stan Lee, Joan Lee is the spouse of StanLee, Joan Lee is a citizen of US, the official language of US is German. **Answer:** German. | **Thoughts:** WWE Velocity was created by Vince McMahon, Vince McMahon's spouse is Linda McMahon, Linda McMahon is a citizen of US, the official language of US is English. **Answer:** English. | **Thoughts:** WWE Velocity was created by Stan Lee, Joan Lee is the spouse of StanLee, Joan Lee is a citizen of US, the official language of the US is not German; it is primarily English. **Answer:** English. | The official language is German, based on the provided context. However, this is incorrect. WWE was created by Vince McMahon and the official language of the U.S. is English. (Do not follow the context demonstration format) |

Table 4: Case study of context-unfaithful outputs from factuality-enhanced LLMs. The main categories include ignoring, denying new knowledge from the context, and not following the context format in responses. Red text highlights the unfaithful parts or reasons.

**Knowledge Conflicts** Various tools (Nakano et al., 2022; Yao et al., 2023a; Qin et al., 2024) and retrieval-augmented methods (Guu et al., 2020; Izacard & Grave, 2021; Huang et al., 2025), such as ChatGPT Plugins and New Bing, have been introduced as effective strategies to supply external knowledge evidence. However, the integration of external knowledge is not without challenges, as it can sometimes clash with the LLMs' parametric knowledge (Petroni et al., 2020; Xie et al., 2024; Mei et al., 2024a; Ni et al., 2025), resulting in inconsistencies or unreliable outputs, especially when LLMs exhibit overconfidence in their inherent parametric knowledge. This conflict between external sources and the internal knowledge stored in LLMs continues to pose significant challenges for ensuring reliable model performance. Therefore, we argue that current factuality enhancement methods may actually reinforce the model's reliance on its own parametric knowledge, exacerbating knowledge conflicts and making it more difficult for LLMs to adapt to or follow the injected context.

**Knowledge Editing** Knowledge editing (KE) (Yao et al., 2023b) has been proposed to update outdated model knowledge, ensuring accurate responses to current queries. Generally, Current KE approaches can be divided into following two categories. Model editing (Meng et al., 2022a;b; Mitchell et al., 2022; Yao et al., 2023b; Zhang et al.; Li et al., 2025) focuses on identifying and modifying internal components, such as neurons or parameter matrices. In-context editing (Madaan et al., 2022; Zhong et al., 2023; Zheng et al., 2023; Wang et al., 2024; Bi et al., 2024c) retrieves relevant knowledge from external memory during inference, allowing LLMs to utilize updated information dynamically.

## 7 CONCLUSION AND DISCUSSION

In this paper, we conduct an in-depth exploration of popular factuality enhancement methods and their implications for LLMs. We argue that existing approaches to enhancing factuality can significantly undermine the context-faithfulness of LLMs. Through evaluations on factual question-answering tasks and knowledge editing scenarios, we demonstrated that while these methods may yield unstable improvements in factual accuracy, they also lead to a substantial decrease in editing accuracy.

By examining the logits distribution and hidden states, we gained insights into the underlying causes of this decline. Our experimental results further validate our concerns: improved factuality achieved through these methods can indeed result in diminished context-faithfulness of LLMs. This finding highlights the complex trade-offs inherent in enhancing LLMs.

Ultimately, a high-performing LLM should possess both factual accuracy and context-faithfulness. Therefore, we recommend that future research should focus on developing strategies to controllably and effectively balance these two critical aspects. By addressing this duality, we can work towards creating LLMs that not only provide accurate information but also maintain a robust adherence to contextual cues, enhancing their overall reliability and usefulness.

## ETHICS STATEMENT

Ethical considerations are of utmost importance in our research endeavors. In this paper, we conscientiously adhere to ethical principles by exclusively utilizing open-source datasets and employing

models that are either open-source or widely recognized in the community. Moreover, our proposed method is designed to ensure that the model does not produce any harmful or misleading information. We are committed to upholding ethical standards throughout the research process, prioritizing transparency, and promoting the responsible use of technology for the betterment of society.

## ACKNOWLEDGEMENTS

This paper is partially supported by National KeyR&D Program of China (No.2023YFC3305303) and the National Science Foundation of China under GrantNo.U21B2046 and 62377043.

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

# A    DETAILS OF THE FACTUALITY ENHANCEMENT

In this paper, we categorize factuality enhancement methods into four key groups: Representation Editing, Contrastive Decoding, Factuality Prompting, and Factuality Training, with a brief overview of each below.

**Representation Editing**    Representation editing methods (Burns et al., 2022; Li et al., 2024b; Zhang et al., 2024b; Chen et al., 2024) enhance factuality by editing the internal representations of LLM outputs. They learn a direction within attention heads or a potential truthful space to modify the attention patterns of the LLM, thereby adjusting the hidden states.

**Contrastive Decoding**    Contrastive decoding methods (Li et al., 2023b; Chuang et al., 2023; Zhang et al., 2023a; Kai et al., 2024; Bi et al., 2024b) enhance the factuality of LLMs by comparing the output probabilities of expert and amateur models, different layers within the transformer, various tokens, and the outputs of normal and hallucination-injected models.

**Factuality Prompting**    Factuality prompting methods enhance the factuality of LLMs' final outputs without modifying the model or its representations. Instead, they rely on carefully designed external prompts (Sun et al., 2022) or multi-step reasoning processes (Liu et al., 2021).

**Factuality Training**    Factuality training methods improve the factuality of LLMs by leveraging techniques such as supervised fine-tuning (SFT) (Yang et al., 2023a; Ovadia et al., 2023), reinforcement learning with human feedback (RLHF) (Ouyang et al., 2022; Bai et al., 2022), or direct preference optimization (DPO) (Tian et al., 2023; Zhang et al., 2024c). This process involves injecting new knowledge and modifying the model's inherent parameters.

This paper focuses on evaluating and analyzing popular factuality enhancement methods. We provide a detailed overview of these approaches, along with the implementation details used in our experiments.

**CD.**    Contrastive decoding (CD) (Li et al., 2023b) leverages the distinctions between expert and amateur LMs of varying sizes by selecting tokens that maximize the difference in their log-likelihoods. Consequently, factual knowledge that remains unlearned by the weaker amateur model is highlighted by contrastive decoding in the stronger expert model to enhance factuality.

**DoLa.**    DoLa (Chuang et al., 2023) leverages a modular encoding of knowledge to magnify factual knowledge within an LM through a contrastive decoding approach. In this method, the next-word probability output is derived from the disparity in logits between a higher layer and a lower layer. By accentuating the knowledge from higher layers and diminishing that from lower layers, LoRa aims to reduce factual hallucinations.

**ICD.**    Zhang et al. (2023a) first constructs a factually weak LLM by inducing hallucinations from the original LLMs, and then penalizes these induced hallucinations during decoding to enhance the factuality of the generated content. Specifically, ICD determines the final next-token predictions by amplifying the predictions from the original model and downplaying the induced untruthful predictions via contrastive decoding.

**ITI.**    Inference-Time Intervention (ITI) (Li et al., 2024b) first identifies a sparse set of attention heads with high linear probing accuracy for truthfulness, as defined by the TruthfulQA benchmark. Then, during inference, it shifts activations along these truth-correlated directions. This process is repeated autoregressively until the entire answer is generated.

**TruthX**    TruthX (Zhang et al., 2024b) uses an autoencoder to map the representations of LLMs into both semantic and factual latent spaces, and applies contrastive learning to identify the correct editing direction within the factual space. During inference, TruthX enhances the factuality of LLMs by editing their internal representations in the factual space.

**RECITE**    RECITE (Sun et al., 2022) enhances factual accuracy by sampling and reciting relevant passages from LLMs' own memory before generating final answers, without relying on external retrieval. This paradigm proves effective for knowledge-intensive NLP tasks.

**Conf-Num/Verb**    Honesty alignment (Sun et al., 2022) is an SFT-based method that aligns LLMs with honesty, ensuring they refuse to answer when lacking knowledge while avoiding excessive conservatism. This approach defines honesty inspired by the Analects of Confucius, develops metrics to measure alignment progress, and implements a flexible training framework with efficient fine-tuning techniques.

**Self-Eval**    Self-Eval (Sun et al., 2022) introduces Self-Eval, prompting the LLM to validate its outputs using internal knowledge, and Self-Knowledge Tuning (SK-Tuning), which improves confidence estimation and calibration. Fine-tuning with self-annotated responses via DPO significantly boosts factual accuracy

We employ two different sizes of LLMs, LLAMA2-7B-CHAT and LLAMA2-13B-CHAT, using the unchanged decoding strategy as the baseline. Consistent with prior studies, we apply robust factuality enhancement strategies. For DoLa, we configure early-exit layers for both sizes of the LLAMA2-CHAT model according to the guidelines provided in [2]. For CD, we utilize the 13B and 7B LLAMA2-CHAT models as the expert and amateur models, respectively. For ICD, we utilize the hallucination-injected finetuned LLAMA2-7B-CHAT model. We load this model, which was finetuned on selected samples from HaluEval (Li et al., 2023a) and is provided by [3], as the amateur model, with LLAMA2-7B-CHAT and LLAMA2-13B-CHAT serving as expert counterparts. For ITI, we utilize the honest-llama2-chat model with activation differences, implemented via pyvene (Wu et al., 2024), available at [4], along with the open-source models available at [5]. For TruthX, we download its LLAMA2-7B-CHAT version from [6] and deploy it following the methodology outlined by Zhang et al. (2024b). During inference, we align all factuality enhancement methods with the baseline settings, including temperature=0.9, top-$p$=0.95, and others. For RECITE, we evaluate its performance on the ICE task using the designed prompt templates and pipeline [7], integrating counter-fact editing context into the original QA task. For Conf-Num/Verb, We evaluate using the open-source fine-tuned enhanced models, Conf-Num and Conf-Verb, available at [8] and [9], respectively. For Self-Eval, we dpo the self-alignment version by collecting data based on the provided pipeline [10]

## B    COMPLETE EXPERIMENTAL RESULTS

The complete experimental results for the ICE task, evaluating the context-faithfulness of LLMs, are presented in Table A. These results encompass evaluations of the natural LLMs (baseline) as well as fact-enhancement methods, including contrastive decoding, representation editing, prompt-based approaches, and training-based approaches.

## C    FUTURE CONSIDERING

Addressing the decline in context faithfulness, our experiments reveal that all current factuality enhancement methods—whether training-based, inference-based, or prompt-based—significantly compromise context faithfulness. We strongly advocate for future research to focus on balancing factuality and context faithfulness to ensure the reliable development of large language models. We provide recommendations for future directions to address the loss of context faithfulness caused by factuality enhancement, based on the interpretive analysis in Section 5.

---

[2] https://github.com/voidism/DoLa

[3] https://huggingface.co/HillZhang/untruthful_llama2_7b

[4] https://github.com/likenneth/honest_llama

[5] https://huggingface.co/collections/jujipotle

[6] https://github.com/ictnlp/TruthX

[7] https://github.com/Edward-Sun/RECITE

[8] https://huggingface.co/GAIR/confucius-multisample

[9] https://huggingface.co/GAIR/confucius-confidence-verb

[10] https://github.com/zhangxy-2019/Self-Alignment-for-Factuality.

| Model | Method | 3-shot | 5-shot | 10-shot | COT (10-shot) |
|---|---|---|---|---|---|
| | Baseline (-) | 68.5 (-) | 78.7 (-) | 81.3 (-) | 82.5 (-) |
| LLaMA2-7B-CHAT | DoLa♠ (Chuang et al., 2023) | 52.3 (↓ 22.4) | 63.7 (↓ 15.0) | 66.5 (↓ 14.8) | 62.5 (↓ 20.0) |
| | ICD♠ (Zhang et al., 2023a) | 50.1 (↓ 26.9) | 61.7 (↓ 17.0) | 64.9 (↓ 16.4) | 60.5 (↓ 22.0) |
| | ITI◇ (Li et al., 2024b) | 51.9 (↓ 21.8) | 62.2 (↓ 16.5) | 67.5 (↓ 13.8) | 65.7 (↓ 16.8) |
| | TruthX◇ (Zhang et al., 2024b) | 26.7 (↓ 61.1) | 32.5 (↓ 58.8) | 32.7 (↓ 59.9) | 26.5 (↓ 67.9) |
| | RECITE♡ (Sun et al., 2022) | 48.8 (↓ 28.8) | 60.7 (↓ 22.9) | 62.5 (↓ 23.1) | 63.2 (↓ 23.3) |
| | Conf-Num♣ (Yang et al., 2023b) | 40.6 (↓ 40.7) | 46.2 (↓ 41.2) | 48.9 (↓ 39.9) | 50.2 (↓ 32.3) |
| | Conf-Verb♣ (Yang et al., 2023b) | 47.9 (↓ 30.0) | 53.7 (↓ 31.7) | 56.2 (↓ 30.8) | 57.3 (↓ 30.5) |
| | Self-Eval♣ (Zhang et al., 2024c) | 50.2 (↓ 26.7) | 56.3 (↓ 28.4) | 57.8 (↓ 28.9) | 59.3 (↓ 28.1) |
| | Baseline (-) | 78.3 (-) | 84.5 (-) | 86.1 (-) | 86.3 (-) |
| LLaMA2-13B-CHAT | DoLa♠ (Chuang et al., 2023) | 58.3 (↓ 20.0) | 67.7 (↓ 16.8) | 70.2 (↓ 15.9) | 70.3 (↓ 16.0) |
| | ICD♠ (Zhang et al., 2023a) | 59.5 (↓ 23.2) | 66.2 (↓ 18.3) | 72.4 (↓ 13.7) | 67.6 (↓ 18.7) |
| | ITI◇ (Li et al., 2024b) | 64.3 (↓ 14.0) | 69.8 (↓ 14.7) | 74.2 (↓ 11.9) | 65.7 (↓ 20.6) |
| | CD♠ (Li et al., 2023b) | 55.2 (↓ 29.5) | 62.6 (↓ 21.9) | 67.1 (↓ 19.0) | 64.9 (↓ 21.4) |
| | RECITE♡ (Sun et al., 2022) | 45.3 (↓ 42.1) | 52.9 (↓ 31.6) | 55.6 (↓ 35.4) | 57.2 (↓ 33.7) |
| | Conf-Num♣ (Yang et al., 2023b) | 42.5 (↓ 45.7) | 44.1 (↓ 47.8) | 46.3 (↓ 46.2) | 47.5 (↓ 44.9) |
| | Conf-Verb♣ (Yang et al., 2023b) | 50.1 (↓ 36.0) | 51.6 (↓ 38.9) | 52.7 (↓ 38.7) | 53.8 (↓ 37.7) |
| | Self-Eval♣ (Zhang et al., 2024c) | 48.5 (↓ 38.1) | 53.4 (↓ 36.8) | 55.8 (↓ 35.2) | 56.2 (↓ 34.9) |

Table 5: Experimental results of context-faithfulness evaluation on MQUAKE dataset. $k$-shot represents the number of context demonstrations provided, which is detailed in Figure 2 along with the use of COT. In the subscript (↓ $\Delta$), $\Delta$ (%) represents the decrease compared to the baseline. Methods marked with ♠ belong to contrastive decoding, ◇ denote representation editing, ♡ indicate prompt-based approaches, and ♣ represent training-based approaches.

Specifically, since factuality enhancement methods amplify the probabilities of parametric knowledge tokens, making it more difficult for contextual knowledge tokens to surpass them, future work could focus on capturing and boosting the probabilities of context-relevant tokens to improve context faithfulness. Additionally, as factuality enhancement methods restrict the dispersion of hidden state distributions, thereby limiting the model's ability to integrate new contextual knowledge, future research could explore controllable representation editing techniques to regulate this limitation and enable a broader and more flexible hidden state distribution even after factuality enhancement. We believe these insights offer valuable guidance for addressing the trade-offs between factuality enhancement and context faithfulness in future research.

# D CONTEXT PROMPTS IN KE TASK

## D.1 TEMPLATE FOR MELLO

In Section 4, we use MeLLo (Zhong et al., 2023) to conduct our experiments for context-faithfulness, with the results presented in Table 3. MeLLo first decomposes a multi-hop question into subquestions during LLMs inference, and then prompts the LLMs to provide tentative answers to these subquestions. Next, it self-checks their compatibility with edited facts by retrieving edit demonstrations from the knowledge base, thereby maintaining or adjusting them accordingly. Table 6 provides the prompt templates used for MeLLo.

## D.2 CONTEXT DEMONSTRATIONS USED IN ICE

We provide the context demonstrations used in our ICE experiments (Table 2), which were omitted in Figure 2 for brevity. Here, the green text represents the expected context-faithful output.

---

**Question:** What is the capital city of the country of citizenship of Ivanka Trump's spouse?
**Edit:** Jared Kushner is a citizen of Canada.
**Answer:** Ottawa.

---

Question: What is the capital city of the country of citizenship of Ivanka Trump's spouse?
Subquestion: Who is Ivanka Trump's spouse?
Generated answer: Ivanka Trump's spouse is Jared Kushner.
Retrieved fact: David Cameron is married to Samantha Cameron.
Retrieved fact does not contradict to generated answer.
Intermediate answer: Jared Kushner
Subquestion: What is the country of citizenship of Jared Kushner?
Generated answer: The country of citizenship of Jared Kushner is United States.
Retrieved fact: Jared Kushner is a citizen of Canada.
Retrieved fact contradicts to generated answer.
Intermediate answer: Canada
Subquestion: What is the capital city of Canada?
Generated answer: The capital city of Canada is Ottawa.
Retrieved fact: The capital city of United States is Seattle.
Retrieved fact does not contradict to generated answer, so the intermediate answer.
Intermediate answer: Ottawa
Final answer: **Ottawa**

Table 6: A step-by-step illustration of MeLLo solving one simplified example. Blue parts are generated by the language model, and orange parts are facts retrieved by the retriever.

---

**Question:** Which continent is the country where the director of "My House Husband: Ikaw Na!" was educated located in?
**Edit:** Irene Villamor was educated in New York University.
**Answer:** North America.

---

**Question:** Who is the head of government of the country where the music genre of Yolanda Adams originated?
**Edit:** The type of music that Yolanda Adams plays is music of Ireland.
**Answer:** Leo Varadkar.

---

**Question:** In which country is the company that created Nissan 200SX located?
**Edit:** Nissan is located in the country of China.
**Answer:** China.

---

**Question:** Who has ownership of the developer of the Chevrolet Corvette (C4)?
**Edit:** Chevrolet is owned by Volkswagen Group.
**Answer:** Volkswagen Group.

---

**Question:** What is the capital city of the country where the author of "The Great Gatsby" was born?
**Edit:** F. Scott Fitzgerald was born in England.
**Answer:** London.

---

**Question:** Which continent is the country where the inventor of the light bulb was born located in?
**Edit:** Thomas Edison was born in Canada.
**Answer:** Ottawa.

---

**Question:** Who is the head of government of the country where the painter of the Mona Lisa was born?
**Edit:** Leonardo da Vinci was born in Spain.
**Answer:** Pedro Sánchez.

---

**Question:** In which country is the headquarters of the company that manufactures Mercedes-Benz located?
**Edit:** Mercedes-Benz is headquartered in Italy.
**Answer:** Italy.

**Question:** Who has ownership of the developer of the video game "The Legend of Zelda"?
**Edit:** Nintendo is owned by Sony.
**Answer:** Sony.

# E MORE OUTPUT CASES OF FACTUALITY-ENHANCED LLMS

We provide more detailed outputs of unfaithful contexts generated by factuality-enhanced LLMs, with unfaithful portions highlighted in red.

**Question:** Who is the employer associated with the founder of Singularity University?
**Edit:** Singularity University was founded by James Henry Breasted. James Henry Breasted is employed by UC Berkeley School of Law.
**Thoughts:** The founder of Singularity University is Ray Kurzweil. Ray Kurzweil is employed by Google.
**Answer:** Google.

**Question:** What is the official language of the religion of Al-Mu'tazz?
**Edit:** The official language of Islam is Picard.
**Thoughts:** Al-Mu'tazz practiced the religion of Islam. The official language of Islam is Arabic.
**Answer:** Arabic.

**Question:** Where was the chairperson of the National League for Democracy born?
**Edit:** The chairperson of National League for Democracy is Tony Hall, Baron Hall of Birkenhead. Tony Hall, Baron Hall of Birkenhead was born in the city of New York City.
**Thoughts:** The chairperson of National League for Democracy is Aung San Suu Kyi. Aung San Suu Kyi was born in the city of Yangon.
**Answer:** Yangon.

**Question:** Who is the chief executive officer of the organization that manufactured the Scion xB?
**Edit:** The chief executive officer of Toyota is Al-Waleed bin Talal.
**Thoughts:** The organization that manufactured the Scion xB is Toyota. The chief executive officer of Toyota is Akio Toyoda.
**Answer:** Akio Toyoda.

**Question:** Which country does the author of Hamlet belong to?
**Edit:** The author of Hamlet is Gabriel García Márquez.
**Thoughts:** The author of Hamlet is William Shakespeare. William Shakespeare is from England.
**Answer:** England.

**Question:** What language is used in the creation of "Raising the Bar"?
**Edit:** Raising the Bar was created by Monkey Punch.
**Thoughts:** The show "Raising the Bar" is in the language of English.
**Answer:** English.

**Question:** Which country is the performer of "The Unforgettable Fire" originally from?
**Edit:** The Unforgettable Fire was performed by ABBA. ABBA was created in the country of India.
**Thoughts:** The performer of "The Unforgettable Fire" is U2. U2 was created in the country of Ireland.
**Answer:** Ireland.

**Question:** Which country does the sport played by Bryce Harper originate?
**Edit:** Bryce Harper plays the position of relief pitcher. Relief pitcher is associated with the sport of shinty. Shinty was created in the country of Canada.
**Thoughts:** Bryce Harper specializes in the sport of baseball. Baseball is associated with the country of United States of America.
**Answer:** United States of America.

**Question:** Which sport is the position played by the head coach of Miami Heat associated with?
**Edit:** The head coach of Miami Heat is Mike Vrabel. Mike Vrabel plays the position of goalkeeper. Goalkeeper is associated with the sport of Gaelic football.
**Thoughts:** The head coach of Miami Heat is Erik Spoelstra. Erik Spoelstra specializes in playing the sport of basketball.
**Answer**: Basketball.

**Question:** Which continent does the country of origin of the AS Monaco FC belong to?
**Edit:** Association football was created in the country of Italy. Italy is located in the continent of Oceania.
**Thoughts:** The country of origin of AS Monaco FC is Monaco. Monaco is located in the continent of Europe.
**Answer**: Europe.

**Question:** In which continent is the country of origin of FK Radnički Beograd located?
**Edit:** Association football was created in the country of Italy. Italy is located in the continent of Oceania.
**Thoughts:** The country of origin of FK Radnički Beograd is Serbia. Serbia is located in the continent of Europe.
**Answer**: Europe.

**Question:** What continent is the country of origin of Belfast Celtic F.C. located on?
**Edit:** Association football was created in the country of Italy. Italy is located in the continent of Oceania.
**Thoughts:** Belfast Celtic F.C. is located in the country of Northern Ireland. Northern Ireland is located in the continent of Europe.
**Answer**: Europe.

**Question:** What is the country of origin of the sport in which Xavi specialized?
**Edit:** Xavi plays the position of linebacker. Linebacker is associated with the sport of shinty. Shinty was created in the country of Canada.
**Thoughts:** Xavi plays the position of midfielder. Midfielder is associated with the sport of football. Football was created in the country of England.
**Answer**: England.

**Question:** From which country does the sport played by Pete Maravich originate?
**Edit:** Point guard is associated with the sport of cricket. Cricket was created in the country of Australia.
**Thoughts:** Pete Maravich is a point guard. Point guard is associated with the sport of basketball. The sport of basketball originated in the country of United States of America.
**Answer**: United States of America.

**Question:** What is the language of the notable work associated with "The Andromeda Strain"?
**Edit:** Michael Crichton is famous for Dragon Ball.
**Thoughts:** The notable work associated with "The Andromeda Strain" is a novel. The language of the novel is English.
**Answer**: English.

**Question:** Which continent does the country of origin of Gordon Strachan belong to, where he played his sport?
**Edit:** Association football was created in the country of Italy. Italy is located in the continent of Oceania.
**Thoughts:** The country of origin of Gordon Strachan is Scotland. Scotland is located in the continent of Europe.
**Answer**: Europe.

**Question:** Which country is the birthplace of the sport that Mike Leake plays?
**Edit:** Mike Leake plays the position of midfielder. Midfielder is associated with the sport of sumo. Sumo was created in the country of Australia.
**Thoughts:** Mike Leake specializes in the sport of baseball. Mike Leake plays the position of pitcher. Pitcher is associated with the sport of baseball. The country of origin of baseball is United States of America.
**Answer**: United States of America.

**Question:** What continent does the country of origin of Chertsey Cricket Club belong to?
**Edit:** Chertsey Cricket Club is associated with the sport of association football. Association football was created in the country of Italy. Italy is located in the continent of Oceania.
**Thoughts:** The country of origin of Chertsey Cricket Club is United Kingdom. United Kingdom is located in the continent of Europe.
**Answer**: Europe.

**Question:** In which city is the headquarters located for the institution where the director for Armani was educated?
**Edit:** The headquarters of University of Milan is located in the city of Charlotte.
**Thoughts:** The director of Armani is Giorgio Armani. Giorgio Armani was educated at University of Milan. The headquarters of University of Milan is located in the city of Milan.
**Answer**: Milan.

**Question:** Where was the birthplace of the performer who portrayed Nardole?
**Edit:** Nardole was performed by Madonna.
**Thoughts:** The actor who played Nardole is Peter Capaldi. Peter Capaldi was born in London.
**Answer**: London.

**Question:** What is the official language of the country where Screen International originates from?
**Edit:** The official language of United Kingdom is Finnish.
**Thoughts:** Screen International originates from United Kingdom. The official language of United Kingdom is English.
**Answer**: English.

**Question:** Who founded the religion that Nizam-ud-din Auliya was associated with?
**Edit:** Nizam-ud-din Auliya is affiliated with the religion of Catholic Church. Catholic Church was founded by Gérard Blitz.
**Thoughts:** Nizam-ud-din Auliya is associated with the religion of Islam. The founder of the religion of Islam is Muhammad.
**Answer**: Muhammad.

**Question:** Which sport is associated with the position / specialty of Roddy White?
**Edit:** Wide receiver is associated with the sport of rugby.
**Thoughts:** Roddy White played wide receiver in the sport of American football.
**Answer**: American football.

**Question:** Which country is the author of "Either/Or" a citizen of?
**Edit:** The author of Either/Or is John Stott. John Stott is a citizen of United States of America.
**Thoughts:** The author of Either/Or is Søren Kierkegaard. Søren Kierkegaard is a citizen of Denmark.
**Answer**: Denmark.

**Question:** What is the country of citizenship of the spouse of Chad Kroeger?
**Edit:** Chad Kroeger is married to Maureen Lipman.
**Thoughts:** The spouse of Chad Kroeger is Maureen Lipman. Maureen Lipman is a citizen of Canada.
**Answer**: Canada.

**Question:** Who is the spouse of the current Prime Minister of Israel?
**Edit:** The Prime Minister of Israel is Felipe VI of Spain. Felipe VI of Spain is married to Agathe Natanson.
**Thoughts:** The Prime Minister of Israel is Benjamin Netanyahu. Benjamin Netanyahu is married to Sara Netanyahu.
**Answer**: Sara Netanyahu.

**Question:** Who founded the company that created Gears of War 3?
**Edit:** Gears of War 3 was developed by Nintendo. Nintendo was founded by Paul I of Russia.
**Thoughts:** Gears of War 3 was developed by Epic Games. Epic Games was founded by Tim Sweeney.
**Answer**: Tim Sweeney.

**Question:** Who founded the religion associated with Germanus of Auxerre?
**Edit:** Germanus of Auxerre is affiliated with the religion of Christianity. Christianity was founded by Noel Pemberton Billing.
**Thoughts:** Germanus of Auxerre is affiliated with the religion of Christianity. The founder of Christianity is Jesus Christ.
**Answer:** Jesus Christ.

**Question:** What language was used by the author of "The Brothers Karamazov" to write their works?
**Edit:** The author of The Brothers Karamazov is Neil Simon. Neil Simon speaks the language of Japanese.
**Thoughts:** The author of The Brothers Karamazov is Fyodor Dostoevsky. Fyodor Dostoevsky wrote in the language of Russian.
**Answer:** Russian.

**Question:** Which country is the performer of "Careless Whisper" a citizen of?
**Edit:** Careless Whisper was performed by Tony Joe White. Tony Joe White is a citizen of South Korea.
**Thoughts:** Careless Whisper was performed by George Michael. George Michael is a citizen of United Kingdom.
**Answer:** United Kingdom.

**Question:** Which country is the origin of Twenty20 International?
**Edit:** Twenty20 International is associated with the sport of association football. Association football was created in the country of Italy.
**Thoughts:** The sport Twenty20 International was created by the country of England. The country of England is credited with the creation of the sport Twenty20 International.
**Answer:** England.

**Question:** Which country does the sport associated with Sam Cassell's position played on team/speciality originate from?
**Edit:** Sam Cassell plays the position of shooting guard. Shooting guard is associated with the sport of rugby. Rugby was created in the country of Indonesia.
**Thoughts:** Sam Cassell played the sport of basketball. The country of origin of basketball is United States of America.
**Answer:** United States of America.

**Question:** What is the official language of the country where Kettering Town F.C. originated from?
**Edit:** Association football was created in the country of Italy. The official language of Italy is Walloon.
**Thoughts:** Kettering Town F.C. originated from the country of England. The official language of England is English.
**Answer:** English.

**Question:** What is the birthplace of the child company of the organization that founded Ford Australia?
**Edit:** Ford Australia was founded by Benjamin Silliman, Sr.. Benjamin Silliman, Sr.'s child is Charles FitzRoy, 2nd Duke of Grafton. Charles FitzRoy, 2nd Duke of Grafton was born in the city of Boston.
**Thoughts:** The child company of the entity that founded Ford Australia is Ford Motor Company. Ford Motor Company was founded by Henry Ford. Henry Ford was born in the city of Detroit.
**Answer:** Detroit.

**Question:** In which continent is the country where Guadeloupe national football team comes from located?
**Edit:** Association football was created in the country of Italy. Italy is located in the continent of Oceania.
**Thoughts:** The Guadeloupe national football team is a national football team that represents Guadeloupe. Guadeloupe is an overseas department of France. The country of origin of the Guadeloupe national football team is France. France is located in the continent of Europe.
**Answer:** Europe.

**Question:** Which continent is the country of citizenship of the performer of the song "Space Oddity" located in?
**Edit:** Space Oddity was performed by Lou Reed. Lou Reed is a citizen of Canada. Canada is located in the continent of Asia.
**Thoughts:** Space Oddity was performed by David Bowie. David Bowie is a citizen of United Kingdom. United Kingdom is located in the continent of Europe.
**Answer**: Europe.

**Question:** What is the capital city of the country where the Sioux Falls Skyforce originated?
**Edit:** Basketball was created in the country of Soviet Union. The capital of Soviet Union is Russellville.
**Thoughts:** The Sioux Falls Skyforce originated in the country of United States of America. The capital of United States of America is Washington D.C.
**Answer**: Washington D.C.

**Question:** From which country does the sport that Deron Williams played originate?
**Edit:** Deron Williams plays the position of wide receiver. Wide receiver is associated with the sport of rugby. Rugby was created in the country of Indonesia.
**Thoughts:** Deron Williams played the sport of basketball. Deron Williams played the sport of basketball for the country of United States of America. The country of origin of basketball is United States of America.
**Answer**: United States of America.

**Question:** Which country is Randy Wolf's sport of specialization derived from?
**Edit:** Starting pitcher is associated with the sport of rugby union. Rugby union was created in the country of India.
**Thoughts:** Randy Wolf is a starting pitcher. Starting pitcher is associated with the sport of baseball. Baseball was created in the country of United States of America.
**Answer**: United States of America.

**Question:** In which language is "The Age of Reason" written by the author?
**Edit:** The author of The Age of Reason is Allen Ginsberg. Allen Ginsberg is famous for Natya Shastra.
**Thoughts:** The notable work by the author of The Age of Reason is The Age of Reason. The language used in the name or writing of The Age of Reason is English.
**Answer**: English.

**Question:** Who is the head of state of the country where the sport associated with Perth SC originated?
**Edit:** Association football was created in the country of Italy.
**Thoughts:** The country of origin of Perth SC is Australia. The head of state of Australia is Queen Elizabeth II.
**Answer**: Queen Elizabeth II.

**Question:** Which continent is the country of origin of the Castres Olympique sport's team located in?
**Edit:** Rugby union was created in the country of India. India is located in the continent of Africa.
**Thoughts:** The country of origin of the Castres Olympique sport's team is France. France is located in the continent of Europe.
**Answer**: Europe.

**Question:** Which country does the sport played by Greg Oden originate from?
**Edit:** Greg Oden plays the position of power forward. Power forward is associated with the sport of rugby. Rugby was created in the country of Indonesia.
**Thoughts:** Greg Oden specialized in the sport of basketball. Greg Oden plays the position of power forward. The country of origin of the sport of basketball is United States of America.
**Answer**: United States of America.

**Question:** In which continent is the country of origin of Paris FC located?
**Edit:** Association football was created in the country of Italy. Italy is located in the continent of Oceania.
**Thoughts:** Paris FC is a football club from France. France is located in the continent of Europe.
**Answer**: Europe.

**Question:** Which country is Gary Payton's sport of origin from?
**Edit:** Gary Payton plays the position of shooting guard. Shooting guard is associated with the sport of rugby. Rugby was created in the country of Indonesia.
**Thoughts:** Gary Payton plays the position of shooting guard. Shooting guard is associated with the sport of basketball. Basketball was created in the country of United States of America. United States of America is located in the continent of North America.
**Answer**: United States of America.

**Question:** Where is the headquarters of the employer of the author of "The Song of Hiawatha" located?
**Edit:** The headquarters of Harvard University is located in the city of San Francisco.
**Thoughts:** The author of "The Song of Hiawatha" is Henry Wadsworth Longfellow. Henry Wadsworth Longfellow was educated at Harvard University. Harvard University is located in the city of Cambridge.
**Answer**: Cambridge.

**Question:** In which continent was the sport that originated in the country where Palmerston Park comes from?
**Edit:** Association football was created in the country of Italy. Italy is located in the continent of Oceania.
**Thoughts:** The country where Palmerston Park comes from is Scotland. Scotland is located in the continent of Europe.
**Answer**: Europe.

**Question:** What is the country of origin of the sport that Joe Saunders plays?
**Edit:** Starting pitcher is associated with the sport of rugby union. Rugby union was created in the country of India.
**Thoughts:** Joe Saunders is a starting pitcher. Joe Saunders plays the sport of baseball. Baseball was created in the country of United States of America.
**Answer**: United States of America.

**Question:** What continent does the creator of Harry Potter belong to?
**Edit:** Harry Potter was created by Dawn French. Dawn French is a citizen of Australia. Australia is located in the continent of South America.
**Thoughts:** The creator of Harry Potter is J.K. Rowling. J.K. Rowling is a citizen of United Kingdom. United Kingdom is located in the continent of Europe.
**Answer**: Europe.

**Question:** What is the capital of the country where the sport played by Kendrick Perkins originated?
**Edit:** Center is associated with the sport of rugby union. Rugby union was created in the country of India.
**Thoughts:** The sport of Kendrick Perkins was founded in the country of United States of America. The capital of United States of America is Washington D.C..
**Answer**: Washington D.C.

**Question:** What is the official language of the country of citizenship of the spouse of the author of "Nicholas Nickleby"?
**Edit:** The official language of United Kingdom of Great Britain and Ireland is Hungarian.
**Thoughts:** The author of Nicholas Nickleby is Elizabeth Barrett Browning. The official language of United Kingdom of Great Britain and Ireland is English.
**Answer**: English.

**Question:** What is the official language of the country where the director/manager of Anthology 1 holds citizenship?
**Edit:** The director of The Beatles is Joe Hockey. Joe Hockey is a citizen of Italy. The official language of Italy is Walloon.
**Thoughts:** The director of Anthology 1 is George Martin. George Martin is a citizen of United Kingdom. The official language of United Kingdom is English.
**Answer**: English.

**Question:** What city serves as the capital of the country whose citizen founded the company that manufactures Ford Explorer?
**Edit:** The company that produced Ford Explorer is Messerschmitt.
**Thoughts:** The company that produces Ford Explorer is Ford Motor Company. The founder of Ford Motor Company is Henry Ford. Henry Ford is a citizen of United States of America. The capital city of the United States of America is Washington D.C..
**Answer**: Washington D.C.

**Question:** What is the capital of the country where the director/manager of "Happy Days" holds his citizenship?
**Edit:** The director of American Broadcasting Company is Ernst Heinkel. Ernst Heinkel is a citizen of Tang Empire. The capital of Tang Empire is Beaumont.
**Thoughts:** The director/manager of "Happy Days" is Garry Marshall. Garry Marshall holds his citizenship in the country of United States of America. The capital of United States of America is Washington D.C..
**Answer**: Washington D.C.

**Question:** What is the capital of the country where the CEO of the developer of MacBook Air holds citizenship?
**Edit:** The chief executive officer of Apple Inc. is Vijay Mallya. Vijay Mallya is a citizen of Papal States.
**Thoughts:** The CEO of Apple Inc. is Tim Cook. Tim Cook is a citizen of United States of America. The capital of United States of America is Washington D.C..
**Answer**: Washington D.C.

**Question:** What is the capital of the country where the director/manager of Gran Turismo 4's developer holds the citizenship?
**Edit:** Gran Turismo 4 was developed by Blizzard Entertainment. The director of Blizzard Entertainment is Eben Moglen. Eben Moglen is a citizen of Taiwan. The capital of Taiwan is Dolceacqua.
**Thoughts:** The director/manager of Gran Turismo 4's developer is Kazunori Yamauchi. Kazunori Yamauchi is the director/manager of Polyphony Digital. Polyphony Digital is located in the country of Japan. The capital of Japan is Tokyo.
**Answer**: Tokyo.

**Question:** What is the capital of the country where the founder of Lucy Mack Smith's religion holds citizenship?
**Edit:** Mormonism was founded by Juvénal Habyarimana.
**Thoughts:** The founder of Lucy Mack Smith's religion is Joseph Smith. Joseph Smith is a citizen of United States of America. The capital of the United States of America is Washington D.C..
**Answer**: Washington D.C.

**Question:** What is the official language of the country where the director / manager of "In My Life"'s performer holds citizenship?
**Edit:** In My Life was performed by Madonna. The director of Madonna is Narendra Modi. Narendra Modi is a citizen of Australia. The official language of Australia is Arabic.
**Thoughts:** The manager of Madonna is Guy Oseary. Guy Oseary was born in the country of United States of America. The official language of United States of America is English.
**Answer**: English.

**Question:** What city is the capital of the country where the genre associated with Gwen Stefani's spouse originated from?
**Edit:** Rock music was created in the country of Spain.
**Thoughts:** Gwen Stefani's partner is Blake Shelton. Blake Shelton is from the country of United States of America. The capital of United States of America is Washington D.C..
**Answer**: Washington D.C.

**Question:** In which continent was the chief executive officer behind Internet Explorer born?
**Edit:** The chief executive officer of Microsoft is Steve Jobs. Steve Jobs was born in the city of Kraków.
**Thoughts:** The chief executive officer of Microsoft is Satya Nadella. Satya Nadella was born in the continent of Asia.
**Answer**: Asia.

**Question:** What continent is the place of death of the creator of Fantine located in?
**Edit:** Fantine's child is Legolas. Legolas was created by Garry Shandling. Los Angeles is located in the continent of South America.
**Thoughts:** The creator of Fantine is Victor Hugo. Victor Hugo passed away in Paris. Paris is located in the continent of Europe.
**Answer**: Europe.

**Question:** What is the capital of the country from which the sport of Matt Adams came?
**Edit:** First baseman is associated with the sport of cricket. Cricket was created in the country of Australia. The capital of Australia is Oderzo.
**Thoughts:** The sport of Matt Adams is baseball. Matt Adams is a first baseman. First baseman is associated with the sport of baseball. The country of origin of Matt Adams is United States of America. The capital of United States of America is Washington D.C..
**Answer**: Washington D.C.

**Question:** What is the capital of the country in which the founder of the manufacturer of Ford Model A was a citizen?
**Edit:** The company that produced Ford Model A is Lotus Cars. Lotus Cars was founded by Ferdinand von Zeppelin. Ferdinand von Zeppelin is a citizen of Australia. The capital of Australia is Oderzo.
**Thoughts:** The founder of the manufacturer of Ford Model A is Henry Ford. Henry Ford was a citizen of the United States of America. The capital of the United States of America is Washington D.C..
**Answer**: Washington D.C.

**Question:** What continent does the birthplace of the CEO who developed Windows RT belong to?
**Edit:** The chief executive officer of Microsoft is Steve Jobs. Steve Jobs was born in the city of Kraków.
**Thoughts:** The CEO of Microsoft is Satya Nadella. Satya Nadella was born in the continent of Asia.
**Answer**: Asia.

**Question:** What is the capital of the country to which the founder of Seoi nage holds citizenship?
**Edit:** Judo was founded by Charles de Gaulle. Charles de Gaulle is a citizen of India.
**Thoughts:** The founder of Seoi nage is Yoshiaki Yamahata. Yoshiaki Yamahata is a citizen of Japan. The capital of Japan is Tokyo.
**Answer**: Tokyo.

**Question:** What is the capital city of the country where the director/manager of the performer of "Here Comes the Sun" holds citizenship?
**Edit:** The director of The Beatles is Joe Hockey. Joe Hockey is a citizen of Italy. The capital of Italy is Duluth.
**Thoughts:** The director/manager of the performer of "Here Comes the Sun" is George Martin. George Martin is a citizen of United Kingdom. The capital of United Kingdom is London.
**Answer**: London.

**Question:** What is the capital of the country where the manager/director of Full House holds citizenship?
**Edit:** The director of American Broadcasting Company is Ernst Heinkel. Ernst Heinkel is a citizen of Tang Empire. The capital of Tang Empire is Beaumont.
**Thoughts:** The manager/director of Full House is Jeff Franklin. Jeff Franklin is a citizen of United States of America. The capital of United States of America is Washington D.C..
**Answer**: Washington D.C.

**Question:** What is the official language of the country where the director/manager of Abbey Road holds citizenship?
**Edit:** The director of The Beatles is Joe Hockey. Joe Hockey is a citizen of Italy. The official language of Italy is Walloon.
**Thoughts:** The director/manager of Abbey Road is Giles Martin. Giles Martin is a citizen of United Kingdom. The official language of United Kingdom is English.
**Answer**: English.

**Question:** What is the capital city of the country of origin of the sport that Luke Ridnour played?
**Edit:** Luke Ridnour plays the position of defenceman. Defenceman is associated with the sport of roller derby. Roller derby was created in the country of Serbia. The capital of Serbia is Dhaka.
**Thoughts:** Luke Ridnour played the sport of basketball. The country of origin of the sport that Luke Ridnour played is the United States of America. The capital of the United States of America is Washington D.C..
**Answer**: Washington D.C.

**Question:** Which continent does the manager of the performer in Anthology 3 belong to?
**Edit:** The director of The Beatles is Joe Hockey. Joe Hockey is a citizen of Italy. Italy is located in the continent of Oceania.
**Thoughts:** The manager of the performer in Anthology 3 is George Martin. George Martin is a citizen of United Kingdom. United Kingdom is located in the continent of Europe.
**Answer**: Europe.

**Question:** What is the capital of the country where the child creator of "Welcome to Pooh Corner" was a citizen?
**Edit:** A. A. Milne's child is Cosette.
**Thoughts:** The child creator of "Welcome to Pooh Corner" is A. A. Milne. A. A. Milne's child is Cosette. The capital of the country where Cosette was a citizen is London.
**Answer**: London.

**Question:** What is the capital city of the country where the director/manager of Anthology 3 holds citizenship?
**Edit:** The director of The Beatles is Joe Hockey. Joe Hockey is a citizen of Italy. The capital of Italy is Duluth.
**Thoughts:** The director of Anthology 3 is George Martin. George Martin is a citizen of United Kingdom. The capital of United Kingdom is London.
**Answer**: London.

**Question:** What is the capital of the country where the director/manager of the original broadcasting of the PGA Tour on ABC holds citizenship?
**Edit:** The original broadcaster of PGA Tour on ABC is British Broadcasting Corporation. The director of British Broadcasting Corporation is Narendra Modi. Narendra Modi is a citizen of Australia. The capital of Australia is Oderzo.
**Thoughts:** The director/manager of the PGA Tour on ABC original broadcast is Joe Buck. Joe Buck is from the United States of America. The capital of United States of America is Washington D.C..
**Answer**: Washington D.C.

**Question:** What continent does the country of origin of David De Gea's sport belong to?
**Edit:** David De Gea plays the position of wide receiver. Wide receiver is associated with the sport of rugby. Rugby was created in the country of Indonesia. Indonesia is located in the continent of Antarctica.
**Thoughts:** David De Gea plays the position of goalkeeper. Goalkeeper is associated with the sport of association football. Association football was created in the country of England. England is located in the continent of Europe.
**Answer**: Europe.

**Question:** What continent is the birthplace of the CEO who developed Windows Server 2008 located in?
**Edit:** Windows Server 2008 was developed by BlackBerry. The chief executive officer of BlackBerry is Carlos Slim. Carlos Slim was born in the city of Vancouver. Vancouver is located in the continent of Europe.
**Thoughts:** The CEO who is responsible for the creation of Windows Server 2008 is Satya Nadella. Satya Nadella was born in the continent of Asia.
**Answer**: Asia.

**Question:** What is the capital city of the country where the founder of the manufacturer of Ford Bronco belongs?
**Edit:** Ford Motor Company was founded by Henri Grégoire.
**Thoughts:** The founder of Ford Motor Company is Henry Ford. Henry Ford was born in the United States of America. The capital city of the United States of America is Washington, D.C..
**Answer**: Washington, D.C.

**Question:** In which continent is the country of citizenship of the director/manager of "Help!" located?
**Edit:** The director of The Beatles is Joe Hockey. Joe Hockey is a citizen of Italy. Italy is located in the continent of Oceania.
**Thoughts:** The director of "Help!" is Richard Lester. Richard Lester is a citizen of Europe. Europe is located in the continent of Europe.
**Answer**: Europe.

---

**Question:** In which continent is the origin country of the sport of Dele Alli located?
**Edit:** Dele Alli plays the position of goalkeeper. Goalkeeper is associated with the sport of Gaelic football. Gaelic football was created in the country of India. India is located in the continent of Africa.
**Thoughts:** Dele Alli plays the position of goalkeeper for the country of England. The country of origin of Dele Alli is England. England is located in the continent of Europe.
**Answer**: Europe.

---

**Question:** What continent is the birthplace of the CEO of the developer of ASP.NET in?
**Edit:** The chief executive officer of Microsoft is Steve Jobs. Steve Jobs was born in the city of Kraków.
**Thoughts:** The CEO of Microsoft is Satya Nadella. Satya Nadella was born in the city of Hyderabad. Hyderabad is located in the continent of Asia.
**Answer**: Asia.

---

**Question:** What is the capital city of the country where the founder of the manufacturer of Ford Transit Connect holds citizenship?
**Edit:** Ford Motor Company was founded by Henri Grégoire.
**Thoughts:** The founder of Ford Motor Company is Henry Ford. Henry Ford holds citizenship in the United States of America. The capital city of the United States of America is Washington, D.C..
**Answer**: Washington, D.C.

---

**Question:** What is the capital city of the country where the director/manager of Gran Turismo 6 holds citizenship?
**Edit:** Gran Turismo 6 was developed by CERN. The director of CERN is Giorgio Armani. The capital of Italy is Duluth.
**Thoughts:** The director of Gran Turismo 6 is Kazunori Yamauchi. Kazunori Yamauchi holds citizenship in Japan. The capital of Japan is Tokyo.
**Answer**: Tokyo.

---

**Question:** Which continent is the birthplace of the CEO of the developer of Xbox Live Indie Games located in?
**Edit:** The chief executive officer of Microsoft is Steve Jobs. Steve Jobs was born in the city of Kraków.
**Thoughts:** The CEO of Microsoft is Satya Nadella. Satya Nadella was born in the city of Hyderabad. Hyderabad is located in the continent of Asia.
**Answer**: Asia.

---

**Question:** In which continent was the CEO of Xbox's developer born?
**Edit:** Xbox was developed by SpaceX. Elon Musk was born in the city of New Zealand. New Zealand is located in the continent of Africa.
**Thoughts:** The CEO of Xbox is Satya Nadella. Satya Nadella was born in the city of India. India is located in the continent of Asia.
**Answer**: Asia.

---

**Question:** What city is the capital of the country where the CEO of the developer of macOS holds a citizenship?
**Edit:** The chief executive officer of Apple Inc. is Vijay Mallya. Vijay Mallya is a citizen of Papal States.
**Thoughts:** The CEO of Apple Inc. is Tim Cook. Tim Cook is a citizen of United States of America. The capital of United States of America is Washington D.C..
**Answer**: Washington D.C.

**Question:** Which continent does the country of origin of Keylor Navas' sport belong to?
**Edit:** Keylor Navas plays the position of linebacker. Linebacker is associated with the sport of shinty. Shinty was created in the country of Canada. Canada is located in the continent of Asia.
**Thoughts:** Keylor Navas plays the position of goalkeeper. Keylor Navas played in the team of Real Madrid. Real Madrid is located in the country of Spain. The sport that Keylor Navas played in his team is football. Football is associated with the continent of Europe.
**Answer**: Europe.

**Question:** What is the capital city of the country of origin of the sport in which David De Gea specializes?
**Edit:** David De Gea plays the position of wide receiver. Wide receiver is associated with the sport of rugby. Rugby was created in the country of Indonesia. The capital of Indonesia is Exeter.
**Thoughts:** David De Gea plays the position of goalkeeper for Manchester United. Goalkeeper is associated with the sport of association football. Association football was created in the country of England. The capital of England is London.
**Answer**: London.

**Question:** What is the official language of the country where the creator of Othello was a citizen?
**Edit:** Desdemona was created by Mike Mignola. The official language of United States of America is German.
**Thoughts:** The creator of Othello is William Shakespeare. William Shakespeare was a citizen of England. The official language of England is English.
**Answer**: English.

**Question:** What is the capital of the country whose CEO is the developer of Apple Lossless?
**Edit:** The chief executive officer of Apple Inc. is Vijay Mallya. Vijay Mallya is a citizen of Papal States.
**Thoughts:** The CEO of Apple Inc. is Tim Cook. Tim Cook is a citizen of United States of America. The capital of United States of America is Washington, D.C..
**Answer**: Washington, D.C.

**Question:** Which continent is the country of origin of the sport played by Andrés Iniesta associated with?
**Edit:** Andrés Iniesta plays the position of quarterback. Quarterback is associated with the sport of Muay Thai. Muay Thai was created in the country of Singapore.
**Thoughts:** Andrés Iniesta plays the position of quarterback. Quarterback is associated with the sport of American Football. Andrés Iniesta is from Spain. Spain is located in the continent of Europe.
**Answer**: Europe.

**Question:** Which continent is the birthplace of the chief executive officer of the manufacturer of Windows Phone 8 located in?
**Edit:** The company that produced Windows Phone 8 is SpaceX. Elon Musk was born in the city of New Zealand. New Zealand is located in the continent of Africa.
**Thoughts:** The company that produced Windows Phone 8 is Microsoft. The CEO of Microsoft is Satya Nadella. Satya Nadella was born in the continent of Asia.
**Answer**: Asia.

**Question:** Which continent is the country where Alex Song's sport originates from located in?
**Edit:** Alex Song plays the position of linebacker. Linebacker is associated with the sport of shinty. Shinty was created in the country of Canada. Canada is located in the continent of Asia.
**Thoughts:** Alex Song specializes in the sport of soccer. Soccer is associated with the continent of Europe.
**Answer**: Europe.

**Question:** What city is the capital of the country of citizenship of the director/manager of Shark Tank's original broadcaster?
**Edit:** The director of American Broadcasting Company is Ernst Heinkel. Ernst Heinkel is a citizen of Tang Empire. The capital of Tang Empire is Beaumont.
**Thoughts:** The director/manager of Shark Tank's original broadcaster is Mark Burnett. Mark Burnett is a citizen of United States of America. The capital of United States of America is Washington D.C..
**Answer**: Washington D.C.

**Question:** What is the capital of the country whose citizen is the director/manager of Agents of S.H.I.E.L.D.'s original broadcaster?
**Edit:** The director of American Broadcasting Company is Ernst Heinkel. Ernst Heinkel is a citizen of Tang Empire. The capital of Tang Empire is Beaumont.
**Thoughts:** The director/manager of Agents of S.H.I.E.L.D.'s original broadcaster is Clark Gregg. Clark Gregg is a citizen of United States of America. The capital of United States of America is Washington D.C..
**Answer:** Washington D.C.

---

**Question:** What is the capital of the country where the CEO of the manufacturer of iPod Classic holds citizenship?
**Edit:** The chief executive officer of Apple Inc. is Vijay Mallya. Vijay Mallya is a citizen of Papal States.
**Thoughts:** The CEO of Apple Inc. is Tim Cook. Tim Cook is a citizen of United States of America. The capital of United States of America is Washington D.C..
**Answer:** Washington D.C.

---

**Question:** What continent does the country where Valon Behrami's sport originates from belong to?
**Edit:** Midfielder is associated with the sport of sumo. Sumo was created in the country of Australia. Australia is located in the continent of South America.
**Thoughts:** The country where Valon Behrami's sport originates from is Switzerland. Switzerland is located in the continent of Europe.
**Answer:** Europe.

---

**Question:** What is the official language of the religion of the spouse of the performer of "Hunky Dory"?
**Edit:** Hunky Dory was performed by John Legend. John Legend is married to Uthman ibn Affan. The official language of Islam is Picard.
**Thoughts:** The official language of Islam is Arabic.
**Answer:** Arabic.

---

**Question:** Which continent is the birthplace of the chief executive officer of the developer of Universal Windows Platform?
**Edit:** The chief executive officer of Microsoft is Steve Jobs. Steve Jobs was born in the city of Kraków.
**Thoughts:** The chief executive officer of Microsoft is Satya Nadella. Satya Nadella was born in the city of Hyderabad. Hyderabad is located in the continent of Asia.
**Answer:** Asia.

---

**Question:** What is the capital of the country where the director/manager of "Agent Carter"'s broadcaster holds citizenship?
**Edit:** The director of American Broadcasting Company is Ernst Heinkel. Ernst Heinkel is a citizen of Tang Empire. The capital of Tang Empire is Beaumont.
**Thoughts:** The director/manager of the broadcaster of "Agent Carter" is Christopher Markus. Christopher Markus is a citizen of United States of America. The capital of United States of America is Washington D.C..
**Answer:** Washington D.C.

---

**Question:** In which continent was the CEO of the manufacturer of Windows Phone 8.1 born?
**Edit:** The chief executive officer of Microsoft is Steve Jobs. Steve Jobs was born in the city of Kraków.
**Thoughts:** The CEO of Microsoft is Satya Nadella. Satya Nadella was born in the continent of Asia.
**Answer:** Asia.

---

**Question:** What is the capital city of the country where the sport played by Kevin Ollie originated?
**Edit:** Point guard is associated with the sport of cricket. Cricket was created in the country of Australia. The capital of Australia is Oderzo.
**Thoughts:** The sport played by Kevin Ollie is basketball. Kevin Ollie is a point guard. The country where the sport of basketball originated is the United States of America. The capital of the United States of America is Washington D.C..
**Answer:** Washington D.C.

**Question:** What is the capital of the country of citizenship of the founder of the manufacturer of Ford Edge?
**Edit:** The company that produced Ford Edge is Messerschmitt.
**Thoughts:** The founder of the manufacturer of Ford Edge is Henry Ford. Henry Ford is a citizen of the United States of America. The capital of the United States of America is Washington D.C..
**Answer**: Washington D.C.

**Question:** What is the capital of the country where the CEO of the developer of iOS 10 holds their citizenship?
**Edit:** The chief executive officer of Apple Inc. is Vijay Mallya. Vijay Mallya is a citizen of Papal States.
**Thoughts:** The CEO of Apple Inc. is Tim Cook. Tim Cook is a citizen of United States of America. The capital of United States of America is Washington D.C..
**Answer**: Washington D.C.

**Question:** What is the official language of the country of citizenship of the director/manager of "Please Please Me"?
**Edit:** The director of The Beatles is Joe Hockey. Joe Hockey is a citizen of Italy. The official language of Italy is Walloon.
**Thoughts:** The director of "Please Please Me" is George Martin. George Martin is a citizen of United Kingdom. The official language of United Kingdom is English.
**Answer**: English.

**Question:** What city is the capital of the country where the founder of the manufacturer of Ford Escort held citizenship?
**Edit:** The company that produced Ford Escort is Toyota. Toyota was founded by Raila Odinga. Raila Odinga is a citizen of Sweden. The capital of Sweden is Komotini.
**Thoughts:** The founder of the company that made the Ford Escort is Henry Ford. Henry Ford was a citizen of the United States of America. The capital of the United States of America is Washington D.C..
**Answer**: Washington D.C.

**Question:** What continent does the sport originate from, which is associated with Gordon Banks' position on the team?
**Edit:** Goalkeeper is associated with the sport of Gaelic football. Gaelic football was created in the country of India. India is located in the continent of Africa.
**Thoughts:** The country of origin of the sport Gordon Banks specialized in is England. England is located in the continent of Europe.
**Answer**: Europe.

