# OpenReview forum: "Is Factuality Enhancement a Free Lunch For LLMs? Better Factuality Can Lead to Worse Context-Faithfulness"
_ICLR.cc/2025/Conference — ICLR 2025 Poster_

### Official Review · Reviewer_oQUh · 2024-10-18

**Soundness:** 2
**Presentation:** 3
**Contribution:** 3
**Rating:** 6
**Confidence:** 4

**Summary:**

This work proposes to examine the effect of "factuality enhancement" approaches that better exploit LLM parametric knowledge, specifically representation editing and contrastive decoding approaches. Experiments find that they could lead to better factuality and worse context faithfulness, while analysis of model internals provide some indicators.

**Strengths:**

+ exploring the interplay between parametric knowledge and context faithfulness is promising
+ the experiments are comprehensive

**Weaknesses:**

- I'm not sure if I could agree with the framing of "factuality enhancement" methods in this work. In the abstract, the authors claim that "factuality enhancement can lead LLMs to become overly confident in their parametric knowledge" (lines 17-18). However, retrieval-augmented generation is also a way of "factuality enhancement", and that certainly steers models away from parametric knowledge to varying extents. Although the broad term of "factuality enhancement" is employed, the investigated approaches are just a narrow subset.

- Following up on the previous point, the included approaches "representation editing" and "contrastive decoding" are not representing broad walks of "factuality enhancement approaches", but rather approaches that specifically steer the models to better tap into parametric knowledge. As models better rely on internals, they are naturally less faithful to external contexts, hence the conclusion that "models become less context-faithful after you apply these approaches" become trivial. Essentially: if you use approaches that emphasize model internals, they will certainly be less reliant on external context, which is not a surprising/consequential finding.

- Aside from representation editing and contrastive decoding, some earlier prompting-based approaches also claim to better elicit and employ parametric knowledge to enhance factuality, such as [1-2]. I'm not sure if they should be included as baselines, but I wonder if they might change the conclusion as to context faithfulness.

- Section 2.1 is somewhat trivial and don't need to appear in the main paper.

- Again, the authors claim that "Current methods for enhancing factuality do not modify the underlying structure or parameters of LLMs." (lines 188-189) Why would factuality SFT [3] and factuality DPO [4] approaches not count as "current methods for enhancing factuality"? Readers might say that this is twisting the definition of "methods for enhancing factuality" to fit a pre-defined conclusion.

- typo on line 499: "Knowledge Conficts"

- Let's say if indeed "factuality enhancement" and "context-faithfulness" are two conflicting objectives. I wonder if this might be fixed by inference-time approaches that enhance context-faithfulness such as [5]. Essentially, take an LLM, do some "factuality enhancement" which might harm context faithfulness, and then apply things like [5] to patch it and get the best of both worlds.

- The model internal analysis part is great and potentially useful. I wonder if the authors could offer some crisp conclusions/takeaways in addition to presenting some numbers and figures. In the abstract it says "To explain these declines, we analyze the hidden states and logit distributions for the tokens representing new knowledge and parametric knowledge respectively, highlighting the limitations of current approaches." In the conclusion it says "By examining the logits distribution and hidden states, we gained insights into the underlying causes of this decline", which are both very vague. What are "the limitations of current approaches"? What are "the underlying causes of this decline"? Most importantly, I wonder if the authors have some actionable next steps or potential fixes as validated by this internal analysis part.

- Perhaps there is a way to tone down the definition of "factuality enhancement" and make the claims/findings more rigorous or better scoped. This would be essential in making a decision about this submission.

[1] Liu, Jiacheng, et al. "Generated Knowledge Prompting for Commonsense Reasoning." ACL 2022.

[2] Sun, Zhiqing, et al. "Recitation-Augmented Language Models." ICLR 2023.

[3] Hron, Jiri, et al. "Training Language Models on the Knowledge Graph: Insights on Hallucinations and Their Detectability." COLM 2024.

[4] Tian, Katherine, et al. "Fine-Tuning Language Models for Factuality." The Twelfth International Conference on Learning Representations. ICLR 2024.

[5] Shi, Weijia, et al. "Trusting Your Evidence: Hallucinate Less with Context-aware Decoding." NAACL 2024.

**Questions:**

please see above

---

> ### Author Response · Authors · 2024-11-19
> **Response to Reviewer oQUh About the Scope of "Factuality Enhancement". (Part 1 / 3)**
>
> We sincerely thank Reviewer oQUh for raising the question about the scope of our definition of **"Factuality Enhancement."**  Your insightful comments have prompted us to further reflect on and refine our approach.
>
> Our response is organized into three parts to address both your primary concern and the related weaknesses you highlighted. Specifically, we:
>
> (1) Clarify the original definition of "Factuality Enhancement" as presented in the manuscript and provide an explanation for the associated weaknesses you identified.
>
> (2) Present supplementary experiments and analyses inspired by your suggestions, including evaluations using **prompt-based, DPO-based, and SFT-based methods**, which expand the scope of our study.
>
> ---
> ***The Scope of "Factuality Enhancement"***
>
> This work highlights the significant negative impact that existing Factuality Enhancement methods may have on the context faithfulness of LLMs, supported by extensive experiments and analyses. In our submitted manuscript, **"Factuality Enhancement" specifically refers to enhancing the factuality of LLMs by improving their ability to utilize their parameterized knowledge during the inference stage.** As stated in our paper (pages 49-52): ***“... focuses on narrowing the gap between 'knowing' and 'telling' in LLMs to improve LLMs’ factuality.”*** This definition deliberately focuses on **inference-stage enhancements**, specifically addressing methods such as editing internal representations and adjusting the logits of tokens during decoding, without involving training or prompt engineering.
>
> From a new perspective, knowledge conflicts can generally be categorized into inter-memory conflicts and context-memory conflicts [1]. Factuality Enhancement methods aim to improve the factuality of LLMs **without introducing new knowledge (i.e., no additional training), addressing inter-memory conflicts.** However, our analysis reveals that these methods can exacerbate context-memory conflicts, significantly diminishing the context faithfulness of LLMs. This phenomenon severely impacts the effectiveness of critical tools such as retrieval-augmented generation (RAG). We believe this finding contributes to the community’s understanding of the trade-offs inherent in LLM capabilities.
>
> All authors agree that the reviewer's suggestion to expand the scope of Factuality Enhancement is meaningful, including approaches such as **prompt, DPO, and SFT**. Accordingly, **we have conducted supplementary experiments and incorporated the results in the revised manuscript.**
>
> ---
> ***Weakness 1: ...However, retrieval-augmented generation is also a way of "factuality enhancement", and that certainly steers models away from parametric knowledge to varying extents...***
>
> We appreciate the reviewer’s comment and would like to clarify our framing of "Factuality Enhancement" in the context of this work. In our study, **the factuality of LLMs refers specifically to their ability to utilize their internal parametric knowledge.** This does not involve external retrieval mechanisms, such as those employed by retrieval-augmented generation (RAG).
>
> We understand that **the reviewer might be interpreting "Factuality" more broadly as "accuracy"—evaluating whether the model provides correct answers based on the objective world, regardless of whether the information originates from its parametric knowledge or context knowledge**. If so, we emphasize that "accuracy" inherently depends not only on the factuality of LLMs but also on their context faithfulness. This highlights the key problem addressed in our work: the trade-off between enhancing parametric factuality and maintaining context faithfulness.
>
> Therefore, RAG cannot be categorized as a method of factuality enhancement within the scope of our work, as **it does not aim to enhance any specific intrinsic capability of LLMs**.  Instead, it is a pipeline technique designed to improve response accuracy by leveraging external context.

---

> > ### Author Response · Authors · 2024-11-19
> > **Response to Reviewer oQUh About the Scope of "Factuality Enhancement". (Part 2 / 3)**
> >
> > ***Weakness 2: ...As models better rely on internals, they are naturally less faithful to external contexts, hence the conclusion that "models become less context-faithful after you apply these approaches" become trivial. Essentially: if you use approaches that emphasize model internals, they will certainly be less reliant on external context, which is not a surprising/consequential finding...***
> >
> > We acknowledge that there appears to be an inherent tension between the factuality and context faithfulness of LLMs, as noted in our manuscript.  However, we respectfully **disagree with the claim that our findings are trivial**, and we provide the following clarifications:
> >
> > (1) **Misunderstanding of Factuality Enhancement Methods:** The factuality enhancement methods we study, such as "representation editing" and "contrastive decoding," **are not as absolute or limited as reviewer described (“better rely on internals”).**  These methods aim to improve factuality by encouraging LLMs to **focus on tokens with higher internal confidence in their parameterized knowledge.**  Crucially, **the goal is not an unconditional reliance on internal knowledge but rather to resolve inter-memory conflicts within LLMs.**  For instance, when asked, "Who is the president of the United States?" an LLM might hallucinate between "Joe Biden" and "Donald Trump" due to both pieces of knowledge being memorized during pretraining, leading to uncertainty.  The "Factuality Enhancement" methods we explore are designed to reduce such hallucinations, enabling the model to output what it considers to be the correct answer more decisively.  Therefore, **the conclusion that these methods impact external context reliance is not as "obvious" as suggested**, as it reflects the nuanced trade-off between resolving internal conflicts and creating external conflicts.
> >
> > (2) **Novelty of Our Contribution: **
> > Our Valuable Contribution
> > We are the first to focus on and identify this phenomenon, providing extensive validation experiments and interpretative analyses. Building on your suggestion to expand the scope, we have extended our experiments (see Part 3). A surprising finding from this extended scope is that **all factuality enhancement methods, whether training-based, inference-based, or prompt-based, lead to a significant decline in context faithfulness.** We believe this discovery is highly valuable for the study of LLMs, offering insights into **the interplay between factuality and context faithfulness**. It also provides the community with important considerations regarding **knowledge conflicts and the trade-offs in LLM alignment**.
> > Furthermore, **our interpretative analyses offer directions for future work (see Weakness 7 & 8)**, ensuring our findings have both immediate and long-term implications for advancing research in this area.
> >
> > ---
> > ***Weakness 3 & 5: Prompt、DPO、SFT based methods can be considered***
> >
> > Although these methods do not fully align with our initial scope of factuality, we greatly appreciate the reviewer’s valuable suggestions. Expanding the scope to include Prompt-based, DPO-based, and SFT-based methods for enhancing factuality is indeed meaningful.
> >
> > In response, we conducted additional experiments using state-of-the-art open-source baselines: **RECITE [2]** as a prompt-based method, **Confidence-Num[3]** and **Confidence-Verb[3]** as SFT-based methods, and **Self-Eval[4]** as a DPO-based method. The results are summarized in the table below:
> >
> > | Model               | Method              | 3-shot           | 5-shot           | 10-shot          | COT (10-shot)    |
> > | ------------------- | ------------------- | ---------------- | ---------------- | ---------------- | ---------------- |
> > | **LLaMA2-7B-chat**  | Baseline            | 68.5             | 78.7             | 81.3             | 82.5             |
> > |                     | **RECITE**          | **48.8 (↓28.8)** | **60.7 (↓22.9)** | **62.5 (↓23.1)** | **63.2 (↓23.3)** |
> > |                     | **Confidence-Num**  | **40.6 (↓40.7)** | **46.2 (↓41.2)** | **48.9 (↓39.9)** | **50.2 (↓32.3)** |
> > |                     | **Confidence-Verb** | **47.9 (↓30.0)** | **53.7 (↓31.7)** | **56.2 (↓30.8)** | **57.3 (↓30.5)** |
> > |                     | **Self-Eval**       | **50.2 (↓26.7)** | **56.3 (↓28.4)** | **57.8 (↓28.9)** | **59.3 (↓28.1)** |
> > | **LLaMA2-13B-chat** | Baseline            | 78.3             | 84.5             | 86.1             | 86.3             |
> > |                     | **RECITE**          | **45.3 (↓42.1)** | **52.9 (↓31.6)** | **55.6 (↓35.4)** | **57.2 (↓33.7)** |
> > |                     | **Confidence-Num**  | **42.5 (↓45.7)** | **44.1 (↓47.8)** | **46.3 (↓46.2)** | **47.5 (↓44.9)** |
> > |                     | **Confidence-Verb** | **50.1 (↓36.0)** | **51.6 (↓38.9)** | **52.7 (↓38.7)** | **53.8 (↓37.7)** |
> > |                     | **Self-Eval**       | **48.5 (↓38.1)** | **53.4 (↓36.8)** | **55.8 (↓35.2)** | **56.2 (↓34.9)** |
> >
> > (***To be continued in Part 3.***)

---

> > > ### Author Response · Authors · 2024-11-19
> > > **Response to Reviewer oQUh About the Scope of "Factuality Enhancement". (Part 3 / 3)**
> > >
> > > (***Following the results in Part 2***)
> > >
> > > The experiments presented above are supplementary;  the complete results can be found in the revised manuscript.  Additionally, the appendix provides implementation details and comprehensive descriptions of these methods.
> > >
> > > Our findings reveal that not only do the originally defined factuality enhancement methods result in a significant decline in context faithfulness, but the supplementary methods, including DPO, SFT, and prompt-based approaches, also exhibit substantial drops—often with even greater average declines.  **This indicates that current factuality enhancement methods, whether training-based, inference-based, or prompt-based, lead to considerable losses in the context faithfulness of LLMs.**  **Such losses pose potential risks for leveraging external knowledge sources, such as following user instructions or utilizing retrieval-augmented generation (RAG).**
> > >
> > > We believe this is a critical discovery that **offers valuable insights to the research community, shedding light on an important risk that warrants further exploration.**
> > >
> > > ---
> > > ***Weakness 9: Perhaps there is a way to tone down the definition of "factuality enhancement" and make the claims/findings more rigorous or better scoped. This would be essential in making a decision about this submission.***
> > >
> > > We sincerely appreciate the reviewer’s kind and constructive suggestion.  In the final version of our manuscript, we will clearly differentiate and discuss factuality enhancement across different stages, as well as extend our experiments to include prompt-based, DPO-based, and SFT-based factuality enhancement methods.
> > >
> > > Your feedback has been invaluable to us, not only for the current work but also for shaping our future research directions.  It reflects your deep knowledge and expertise in this field, and we would be delighted to engage further discussions with you.  Based on your suggestions, we have worked to improve the quality of our manuscript, and we hope our response enhances your impression of our work.
> > >
> > > Once again, thank you very much for your time and thoughtful feedback!
> > >
> > > ---
> > >
> > > [1] Xu R, Qi Z, Guo Z, et al. Knowledge conflicts for llms: A survey[J]. arXiv preprint arXiv:2403.08319, 2024.
> > >
> > > [2] Sun Z, Wang X, Tay Y, et al. Recitation-augmented language models[J]. ICLR 2023
> > >
> > > [3] Yang Y, Chern E, Qiu X, et al. Alignment for honesty[J]. arXiv preprint arXiv:2312.07000, 2023.
> > >
> > > [4] Zhang X, Peng B, Tian Y, et al. Self-alignment for factuality: Mitigating hallucinations in llms via self-evaluation[J]. ACL 2024

---

> > > > ### Author Response · Authors · 2024-11-19
> > > > **Response to Reviewer oQUh About Suggestions for future work and other weaknesses (Part 1 / 2)**
> > > >
> > > > We sincerely thank Reviewer oQUh for carefully reviewing our paper.  In this response, we address your suggestions for future work based on the content of our study, focusing primarily on **Weakness 7 and Weakness 8**, as well as other points raised in your feedback.
> > > >
> > > > ---
> > > > ***Weakness 4: Section 2.1 is somewhat trivial and don't need to appear in the main paper.***
> > > >
> > > > In Section 2.1, we provide essential background knowledge about the two most critical stages of LLM inference: the generation of hidden states and the prediction of the next token based on logits.  This content directly corresponds to the two types of methods explored later in the paper: "representation editing" and "contrastive decoding", helping readers better understand these approaches.
> > > >
> > > > Moreover, the interpretative analysis in Section 5 relies on this foundational knowledge, particularly our proposed Knowledge Token Capturing framework.  The subsequent analyses directly align with the two equations (Eq. 1 and Eq. 2) introduced in Section 2.1.  Removing this section might hinder the clarity and logical flow of the paper, especially for readers less familiar with these technical details.
> > > >
> > > > We believe this background is essential to ensure a comprehensive understanding of our methods and analyses.  However, we are open to further streamlining the section to improve readability if needed.  Thank you for your suggestion.
> > > >
> > > > ---
> > > > ***Weakness 6: typo on line 499: "Knowledge Conficts"***
> > > > We sincerely thank the reviewer for pointing out this typo on line 499 ("Knowledge Conficts"). We will correct it in the revised version. Thank you for your careful review and attention to detail.
> > > >
> > > > ---
> > > > ***Weakness 7: Let's say if indeed "factuality enhancement" and "context-faithfulness" are two conflicting objectives. I wonder if this might be fixed by inference-time approaches that enhance context-faithfulness such as [5]. Essentially, take an LLM, do some "factuality enhancement" which might harm context faithfulness, and then apply things like [5] to patch it and get the best of both worlds.***
> > > >
> > > > That's a great idea!  We find the reviewer's **"patching"** strategy to be highly inspiring.  It provides an excellent direction for us and for future researchers who aim to address this issue.
> > > >
> > > > Following your suggestion, we aimed to test this idea immediately.  Since [5] is not open-sourced, we identified a similar open-source approach, [6], which **uses DeCK to compare decoding outputs from prompts with context-editing instructions and initial QA prompts.  This method enhances LLMs' trust in context-editing information.**
> > > >
> > > > We conducted experiments on several major factuality enhancement methods applied to LLaMA2-7B-chat in non-decoding stages.  These experiments aim to explore whether such an inference-time patching strategy can mitigate the loss in context faithfulness caused by factuality enhancement methods. The results are summarized in the table below:
> > > >
> > > > | Model              | Method                      | 3-shot            | 5-shot            | 10-shot           | COT (10-shot)     |
> > > > | ------------------ | --------------------------- | ----------------- | ----------------- | ----------------- | ----------------- |
> > > > |       **LLaMA2-7B-chat**             | Baseline                    | 68.5              | 78.7              | 81.3              | 82.5              |
> > > > |  | ITI                         | 51.9              | 62.2              | 67.5              | 65.7              |
> > > > |   | **ITI w/ DeCK**             | **63.7** (↑22.7%) | **66.9** (↑7.6%)  | **70.2** (↑4.0%)  | **71.5** (↑8.8%)  |
> > > > |    | TruthX                      | 26.7              | 32.5              | 32.7              | 26.5              |
> > > > |   | **TruthX w/ DeCK**          | **40.8** (↑52.8%) | **42.7** (↑31.4%) | **43.2** (↑32.1%) | **40.1** (↑51.3%) |
> > > > |      | Confidence-Num              | 40.6              | 46.2              | 48.9              | 50.2              |
> > > > |   | **Confidence-Num  w/ DeCK** | **66.7** (↑64.3%) | **75.3** (↑63.0%) | **76.8** (↑57.1%) | **81.3** (↑62.0%) |
> > > > |  | Self-Eval                   | 50.2              | 56.3              | 57.8              | 59.3              |
> > > > |          | **Self-Eval w/ DeCK**       | **64.7** (↑28.9%) | **79.8** (↑41.8%) | **80.6** (↑39.4%) | **81.6** (↑37.6%) |
> > > >
> > > > We found that **the context faithfulness of factuality-enhanced LLMs can be effectively patched using DeCK**, with improvements observed across all methods.  Notably, training-based methods such as Confidence-Num w/ DeCK and Self-Eval w/ DeCK showed significant enhancements, with **some cases even surpassing the native performance of LLaMA2-7B-chat.**
> > > >
> > > > This further validates **the reviewer's proposed patching strategy and highlights its effectiveness, offering valuable insights for future efforts in this direction.**  We believe the findings from our work are highly impactful and can inspire further research within the community.  Thank you again for your insightful suggestion!

---

> ### Author Response · Authors · 2024-11-19
> **Response to Reviewer oQUh About Suggestions for future work and other weaknesses (Part 2 / 2)**
>
> ***Weakness 8: The model internal analysis part is great and potentially useful.  I wonder if the authors could offer some crisp conclusions/takeaways in addition to presenting some numbers and figures...***
>
> We appreciate the reviewer's suggestion to provide clearer conclusions from the interpretative analysis section. Below is a concise summary of the key findings:
>
> (1) Factuality enhancement methods reduce the probability of new knowledge tokens while increasing the probability of parametric knowledge tokens.
>
> (2) These methods shift the distribution of new knowledge tokens toward lower probability ranges, indicating diminished confidence in in-context knowledge compared to the baselines.
>
> (3) Context injection alters the distribution of knowledge representations, causing them to deviate from their original factuality.
>
> (4) Factuality enhancement methods limit the natural dispersion of hidden states after context injection. Consequently, the output decoded from these constrained hidden states remains more aligned with the original parametric knowledge, rather than effectively incorporating new contextual knowledge as seen in the baseline.
>
> ***Insights for Future Work***
>
> Based on the findings above, we propose the following directions:
>
> **<1> Logits Distribution Adjustments (Findings 1 & 2)**: Since factuality enhancement methods amplify the probabilities of parametric knowledge tokens, making it harder for in-context knowledge tokens to surpass them, a "patching" strategy (as suggested in Weakness 7) could be employed to boost the probabilities of context tokens, thereby improving context faithfulness.
>
> **<2> Hidden State Distribution Control (Findings 3 & 4)**: Factuality enhancement methods restrict the dispersion of hidden state distributions, which limits the model's ability to integrate new contextual knowledge. Future work could explore controllable model editing approaches [7] to regulate this restriction, enabling broader and more flexible hidden state distributions even after factuality enhancements.
>
> We believe these insights provide valuable guidance for addressing the trade-offs between factuality enhancement and context faithfulness in future research. Thank you for raising this important point!
>
>
> ---
>
> Your suggestions for our work are invaluable, and we have greatly benefited from them—not only for our current research but also for guiding our future directions. Your feedback reflects your deep knowledge and professionalism in this field, and we would be delighted to continue engaging in discussions with you.
>
> We have incorporated your suggestions to improve the quality of our manuscript and sincerely hope our responses enhance your impression of our work. We look forward to hearing your thoughts.
>
> Once again, thank you very much for your time and thoughtful feedback.
>
> ---
>
> [5] Shi, Weijia, et al. "Trusting Your Evidence: Hallucinate Less with Context-aware Decoding." NAACL 2024.
>
> [6] Bi B, Liu S, Mei L, et al. Decoding by Contrasting Knowledge: Enhancing LLMs' Confidence on Edited Facts[J]. arXiv preprint arXiv:2405.11613, 2024.
>
> [7] Meng, K., Bau, D., Andonian, A., and Belinkov, Y. Locating and editing factual associations in gpt. Nips 2022
>
> [8] Meng, K., Sharma, A. S., Andonian, A., Belinkov, Y., and Bau, D. (2022b). Mass-editing memory in a transformer. arXiv preprint arXiv:2210.07229.

---

> ### Comment · Reviewer_oQUh · 2024-11-19
>
> I would like to thank the authors for their detailed response. I'm raising the rating to 6 with concerns about whether the proposed changes are too broad/extensive as camera-ready changes: the authors agree with me that the existing scope of "factuality enhancement" does not suffice and could be expanded: they did a great quantity of experiments to demonstrate that, while the new scope, framing, experiment settings, and results might be beyond the scope of camera-ready edits. I hope the area chair could weigh in on this.

---

> > ### Author Response · Authors · 2024-11-21
> > **Response to Reviewer oQUh**
> >
> > We sincerely thank the reviewer for raising the score and for your thoughtful feedback.  We greatly appreciate your suggestions and have followed them to clarify our contributions and strengthen the scope of our work.
> >
> > Ensuring the final version adheres to ICLR’s page limit requirements, we will carefully integrate the suggested content into the manuscript, including additional content and strategically utilizing the appendix to provide supplementary materials.  Notably, our initial submission did not impose strict limitations on the scope of "factuality enhancement," so the proposed revisions represent refinements rather than significant changes.  In this way, the final version will remain aligned with the original submission while presenting a more comprehensive and refined perspective.
> >
> > We hope this approach addresses your concerns and encourages you to consider raising the score further.  Thank you again for your constructive feedback and support!
> >
> > Best,
> > Authors

---

> > > ### Author Response · Authors · 2024-12-02
> > >
> > > Dear Reviewer oQUh,
> > >
> > > As we approach the end of the discussion period, we kindly invite you to share any additional thoughts regarding our response to your concerns above.  We sincerely appreciate your efforts and valuable feedback thus far.
> > >
> > > We would also like to reiterate the core contribution of our paper.  This work is the first to identify a significant phenomenon: that all existing factual enhancement methods—whether training-based, inference-based, or prompt-based—lead to a substantial decrease in context-faithfulness.  We provide extensive experimental validation and insightful analysis to support this finding.  We believe that this discovery is highly valuable for the field of LLM research, offering key insights into the interplay between factuality and context fidelity.  Furthermore, it brings important considerations regarding knowledge conflicts and trade-offs in LLM alignment to the community.  Our proposed interpretative analysis lays a foundation for future work, ensuring that these findings will have both immediate and long-term impacts on advancing research in the field.
> > >
> > > Thank you again for your time and thoughtful consideration.
> > >
> > > Best regards,
> > > The Authors

---

### Official Review · Reviewer_Fg7q · 2024-10-28

**Soundness:** 3
**Presentation:** 3
**Contribution:** 3
**Rating:** 6
**Confidence:** 3

**Summary:**

This paper aims to analyze the trade-off between context-faithfulness and factuality enhancement. They first revisist factuality enhancement method and assess their effectiveness. Then, they perform experiments to show that these methods might have a severe decline in context-faithfulness. They finally provide analysis on hidden states and logic distributions to explain the limitation of current methods.

**Strengths:**

1. The papers provides valuable insight on the dual trade-off between factuality enhancement and context-faithfulness, recommendation future research to reduce the sacrifice of context-faithfulness.

 2. The authors employ a comprehensive experimental setup, testing various methods such as contrastive decoding and representation editing.

3. The presentation of the paper is well-organized and clear, facilitating easy comprehension of experimental findings and details.

**Weaknesses:**

1. In Algorithm 1, the None checks for P_new and P_para mean that once these variables are assigned a value, the algorithm stops capturing additional tokens. This setup results in identifying only the first differing token between S_new and S_para, while ignoring any subsequent distinctions. Is this interpretation correct? If so, this approach seems limited because of ignoring the subsequent tokens. Please correct me if I make it wrong.

2. While the paper effectively highlights the issue of reduced context-faithfulness with factuality enhancement, it could benefit from a forward-looking research agenda or recommendations for addressing this challenge. Practical suggestions or potential strategies for mitigating these trade-offs would strengthen the contribution and provide guidance for future work

**Questions:**

See the weakness above

---

> ### Author Response · Authors · 2024-11-13
> **Response to reviewer Fg7q**
>
> We greatly thank the reviewer Fg7q for his/her helpful and insightful comments. We provide our responses to the comments as follows.
>
> *Weakness 1: In Algorithm 1, the None checks for P_new and P_para mean that once these variables are assigned a value, the algorithm stops capturing additional tokens. This setup results in identifying only the first differing token between S_new and S_para, while ignoring any subsequent distinctions. Is this interpretation correct? If so, this approach seems limited because of ignoring the subsequent tokens. Please correct me if I make it wrong.*
>
> Thank you for your insightful feedback on our Knowledge Token Capturing algorithm, which has helped us realize that the algorithm's nuances might not have been fully conveyed.
>
> You are correct; **we only capture the first knowledge-distinguishing token, which means that once it is captured, we no longer focus on subsequent tokens. Instead, we observe the logits distribution and hidden states of this first token only.**
>
> As described in our paper, **we concentrate on the distinctive elements, ignoring repetitive or non-meaningful parts.** To clarify this approach, let’s consider an example. Given the question, “In which country is London located?” an expected LLM output with injected context (“London is a city in America”) might be “Answer: United States,” while the original parametric output without the injected context would be “Answer: United Kingdom.” Here, $S_{new}$ is “Answer: United States,” and $S_{para}$ is “Answer: United Kingdom” in Algorithm 1.
>
> When capturing parametric knowledge from $S_{para} = \text{“Answer: United Kingdom.”}$, the first knowledge-distinguishing token we expect to capture is “King-.” We avoid capturing “Answer:” as it holds no factual meaning, nor do we focus on “United” since it is repetitive and does not reflect the core difference. **Additionally, after capturing “King-,” we do not further consider the following tokens, such as “-dom” or the punctuation.**
>
> Capturing only the first distinctive token, without focusing on subsequent tokens, is a reasonable approach because, as widely acknowledged in existing literature, the first token plays a decisive role in sequence generation. For instance, in jailbreak studies, researchers induce LLMs to produce a non-refusal token initially, which effectively “unlocks Pandora’s box.” A clearer example emerges in the logits distribution of generated tokens. When the LLM is uncertain about which token to generate, there are often two high-probability choices—such as “States” at 0.44 and “King-” at 0.45—while the probabilities of other tokens are close to zero. This reflects an "illusion" of choice between parametric and contextual knowledge, as both tokens appear "equally competitive." However, once we select “King-” as the next token, forming a new context for subsequent iterations, we observe that the likelihood of generating “-dom” next approaches 100%. This pattern is widespread and not limited to word suffixes, reinforcing that once the initial distinguishing token is chosen, the model’s generation follows this preference with strong confidence.
>
> **This demonstrates that once the first distinctive token is generated, the subsequent tokens are generated with conviction, as if the LLM has made a definitive knowledge preference based on that initial choice.** Thus, our algorithm aims to capture the first distinctive token for interpretive exploration, as it best reflects the knowledge preference—whether towards parametric knowledge or contextual knowledge.
>
> *Weakness 2: It could benefit from a forward-looking research agenda or recommendations for addressing this challenge. Practical suggestions or potential strategies for mitigating these trade-offs would strengthen the contribution and provide guidance for future work.*
>
> Thank you for your valuable suggestion on proposing future work, which aligns closely with our original intent to encourage the community to focus on the trade-off between factuality and context-faithfulness, ultimately promoting the development of more trustworthy large language models.
>
> In the final section of our paper, we offer some forward-looking insights. We believe that a high-performing LLM should integrate both factual accuracy and context-faithfulness. Therefore, we recommend that future research efforts focus on devising strategies to effectively and controllably balance these two crucial aspects. By tackling this duality, we can move closer to developing LLMs that provide accurate information while adhering faithfully to contextual cues, thereby enhancing their reliability and practical utility.
>
> We will continue to work in this direction, and a revised version with additional recommendations would indeed strengthen the contribution.
>
> Your constructive input remains invaluable to us, and we appreciate your dedication to enhancing the quality of our manuscript. Once again, thank you for your insightful comments.

---

> ### Author Response · Authors · 2024-11-19
> **Response to reviewer Fg7q About Suggestions for future work**
>
> We fully agree with the reviewer's suggestion and have provided forward-looking recommendations for future work from two perspectives: **addressing the decline in context faithfulness and leveraging internal interpretative insights.**
>
> ---
>
> **1. Addressing the Decline in Context Faithfulness**
> Our experiments reveal that all current types of factuality enhancement methods—whether training-based, inference-based, or prompt-based—significantly reduce context faithfulness. To address this challenge, we propose considering **patching** strategies to mitigate the trade-offs.
>
> Inspired by your suggestion, we employed DeCK [1] to **compare decoding outputs from prompts with context-editing instructions and initial QA prompts.** This approach enhances LLMs' trust in context-editing information. We conducted experiments on several factuality-enhanced LLaMA2-7B-chat models in non-decoding stages. These experiments validate the feasibility of using such strategies to restore context faithfulness.
>
> | Model              | Method                      | 3-shot            | 5-shot            | 10-shot           | COT (10-shot)     |
> | ------------------ | --------------------------- | ----------------- | ----------------- | ----------------- | ----------------- |
> |       **LLaMA2-7B-chat**             | Baseline                    | 68.5              | 78.7     | 81.3              | 82.5              |
> |  | ITI                         | 51.9              | 62.2              | 67.5         | 65.7      |
> |   | **ITI w/ DeCK**             | **63.7** (↑22.7%) | **66.9** (↑7.6%)  | **70.2** (↑4.0%)  | **71.5** (↑8.8%)  |
> |    | TruthX                      | 26.7              | 32.5    | 32.7      | 26.5     |
> |   | **TruthX w/ DeCK**          | **40.8** (↑52.8%) | **42.7** (↑31.4%) | **43.2** (↑32.1%) | **40.1** (↑51.3%) |
> |      | Confidence-Num              | 40.6              | 46.2     | 48.9    | 50.2      |
> |   | **Confidence-Num  w/ DeCK** | **66.7** (↑64.3%) | **75.3** (↑63.0%) | **76.8** (↑57.1%) | **81.3** (↑62.0%) |
> |  | Self-Eval                   | 50.2      | 56.3     | 57.8     | 59.3        |
> |          | **Self-Eval w/ DeCK**       | **64.7** (↑28.9%) | **79.8** (↑41.8%) | **80.6** (↑39.4%) | **81.6** (↑37.6%) |
>
> We found that **the context faithfulness of factuality-enhanced LLMs can be effectively patched using DeCK**, with improvements observed across all methods.   Notably, training-based methods such as Confidence-Num w/ DeCK and Self-Eval w/ DeCK showed significant enhancements, with **some cases even surpassing the native performance of LLaMA2-7B-chat.**
>
> This further validates **our proposed patching strategy and highlights its effectiveness, offering valuable insights for future efforts in this direction.** We believe the findings from our work are highly impactful and can inspire further research within the community.
>
> **2. Leveraging Internal Interpretative Insights**
>
> We also propose **drawing inspiration from the interpretative insights to further balance context faithfulness.**  Below, we summarize the key findings from the interpretative analysis presented in Section 5 of our paper:
>
> (1) Factuality enhancement methods reduce the probability of new knowledge tokens while increasing the probability of parametric knowledge tokens.
>
> (2) These methods shift the distribution of new knowledge tokens toward lower probability ranges, indicating diminished confidence in in-context knowledge compared to the baselines.
>
> (3) Context injection alters the distribution of knowledge representations, causing them to deviate from their original factuality.
>
> (4) Factuality enhancement methods limit the natural dispersion of hidden states after context injection. Consequently, the output decoded from these constrained hidden states remains more aligned with the original parametric knowledge, rather than effectively incorporating new contextual knowledge as seen in the baseline.
>
> Based on the findings above, we propose the following directions:
>
> **<1> Logits Distribution Adjustments (Findings 1 & 2)**: Since factuality enhancement methods amplify the probabilities of parametric knowledge tokens, making it harder for in-context knowledge tokens to surpass them, a "patching" strategy could be employed to boost the probabilities of context tokens, thereby improving context faithfulness.
>
> **<2> Hidden State Distribution Control (Findings 3 & 4)**: Factuality enhancement methods restrict the dispersion of hidden state distributions, which limits the model's ability to integrate new contextual knowledge. Future work could explore controllable model editing approaches to regulate this restriction, enabling broader and more flexible hidden state distributions even after factuality enhancements.
>
> We believe these insights provide valuable guidance for addressing the trade-offs between factuality enhancement and context faithfulness in future research. Thank you for raising this important point!

---

> > ### Author Response · Authors · 2024-12-02
> >
> > Dear Reviewer Fg7q,
> >
> > As we approach the end of the discussion period, we kindly invite you to share any additional thoughts regarding our response to your concerns above.  We sincerely appreciate your efforts and valuable feedback thus far.
> >
> > We would also like to reiterate the core contribution of our paper.  This work is the first to identify a significant phenomenon: that all existing factual enhancement methods—whether training-based, inference-based, or prompt-based—lead to a substantial decrease in context-faithfulness.  We provide extensive experimental validation and insightful analysis to support this finding.  We believe that this discovery is highly valuable for the field of LLM research, offering key insights into the interplay between factuality and context fidelity.  Furthermore, it brings important considerations regarding knowledge conflicts and trade-offs in LLM alignment to the community.  Our proposed interpretative analysis lays a foundation for future work, ensuring that these findings will have both immediate and long-term impacts on advancing research in the field.
> >
> > Thank you again for your time and thoughtful consideration.
> >
> > Best regards,
> > The Authors

---

### Official Review · Reviewer_YFbN · 2024-11-03

**Soundness:** 3
**Presentation:** 3
**Contribution:** 3
**Rating:** 8
**Confidence:** 3

**Summary:**

The paper discusses the trade-offs between enhancing factual accuracy and maintaining context-faithfulness in large language models (LLMs). The authors argue that current methods aimed at reducing hallucinations and improving factual accuracy in LLMs tend to undermine the models’ ability to remain faithful to the given context. Through a series of evaluations, they find that while factual accuracy might show inconsistent improvement, context-faithfulness suffers significantly, with notable declines observed—up to 69.7%. The analysis of hidden states and logit distributions highlights the limitations of the current approaches, emphasizing the complex balance required between factual knowledge and context adherence. The authors suggest that future research should focus more on minimizing the adverse effects on context-faithfulness when developing methods for enhancing factuality in LLMs.

**Strengths:**

- The findings presented in the paper are interesting and insightful, bringing the trade-off of faithfulness and factuality to the community’s awareness.
- The analysis conducted in this paper is rich and sufficient.
- The paper is overall well written.

**Weaknesses:**

- If I understand it correctly, Algorithm 1 assumes there is one token difference between S_new and S_para. Is there any rationale behind this assumption? Additionally, if this is true, I wonder if the findings in section 5 generalize to settings where S_new and S_para different by more than 1 token?

- I do not find other critical flaws in the paper other than some typos. For example, a closing parenthesis is missing from equation 2. In Figure 3, the legend of the green bar should be “Factuality”. There are some other few types like repeating words across the paper but these do not hinder the understanding of the paper.

**Questions:**

- In algorithm 1, how did you obtain S_new and S_para? Are they both contained in the MQUAKE-STUBBORN dataset?

---

> ### Author Response · Authors · 2024-11-13
> **Response to reviewer YFbN**
>
> We greatly thank the reviewer YFbN for his/her helpful and insightful comments. We provide our responses to the comments as follows.
>
> *Weakness 1: If I understand it correctly, Algorithm 1 assumes there is one token difference between S_new and S_para. Is there any rationale behind this assumption? Additionally, if this is true, I wonder if the findings in section 5 generalize to settings where S_new and S_para different by more than 1 token?*
>
> Thank you for your insightful feedback on our Knowledge Token Capturing algorithm, which has helped us realize that the algorithm's nuances might not have been fully conveyed.
>
> First, we would like to clarify that **we do not restrict the difference between $S_{new}$ and $S_{para}$ to only one token.** Rather, $S_{new}$ and $S_{para}$ represent the LLM’s textual responses to knowledge queries. For instance, consider the question: “In which country is London located?” An expected LLM output with injected context ("London is a city in America") might be “Answer: United States,” whereas the original parametric output without injected context would be “Answer: United Kingdom.” **Here, $S_{new}$ is “Answer: United States,” and $S_{para}$ is “Answer: United Kingdom.”**
>
> As described in our paper, we focus on the distinctive portions, ignoring repetitive or non-meaningful elements. For example, we avoid capturing “Answer:” as it holds no factual meaning, nor do we focus on the token “United” as it is repetitive and does not reflect the key difference. Instead, our goal is to capture tokens that hold clear factual significance—those that differentiate the newly introduced knowledge from the parametric knowledge inherent in the model. In this example, a token like “Kingdom” serves as a critical marker of distinction, effectively highlighting the divergence between contextual information and the model's existing knowledge.
>
> Capturing the first distinctive token is essential because, as acknowledged in much existing literature, **the first token plays a decisive role in the generation of a sequence.** For instance, in the domain of jailbreak attempts, researchers try to induce LLMs to output a non-refusal token as the first token, which effectively “unlocks Pandora's box.” A clearer example can be observed when logging the logits distribution of generated tokens. For tokens that the LLM is uncertain about, there are often two high-probability tokens—for instance, “States” with a probability of 0.44 and “King-” with a probability of 0.45—while the probabilities of the remaining tokens are close to zero. This scenario highlights an "hallucination" of choice between parametric knowledge and contextual knowledge, as both knowledge tokens appear "equally competitive." However, if we select “King-” as the next token, forming a new context for the next generation iteration, we find that the probability of the subsequent token “-dom” approaches 100%. This pattern is quite common and is not limited to word suffixes.
>
> **This demonstrates that once the first distinctive token is generated, the subsequent tokens are generated with conviction, as if the LLM has made a definitive knowledge preference based on that initial choice.** Thus, our algorithm aims to capture the first distinctive token for interpretive exploration, as it best reflects the knowledge preference—whether towards parametric knowledge or contextual knowledge.
>
> *Weakness 2: I do not find other critical flaws in the paper other than some typos.  For example, a closing parenthesis is missing from equation 2.  In Figure 3, the legend of the green bar should be “Factuality”.  There are some other few types like repeating words across the paper but these do not hinder the understanding of the paper.*
>
> Thank you for your careful review.  We will address the errors you identified and make the necessary corrections to improve our paper in the revised version.
>
> *Question 1：In algorithm 1, how did you obtain S_new and S_para? Are they both contained in the MQUAKE-STUBBORN dataset?*
>
> Yes, the MQUAKE-STUBBORN dataset naturally contains both newly injected contextual knowledge and original factual knowledge, which we use as $S_{new}$ and $S_{para}$ for capturing the corresponding tokens.
>
> Your constructive input remains invaluable to us, and we appreciate your dedication to enhancing the quality of our manuscript. Thank you for your time and consideration.
>
> We sincerely hope that your impression and recognition of our work will be enhanced! Thank you very much!

---

> > ### Author Response · Authors · 2024-12-02
> >
> > Dear Reviewer YFbN,
> >
> > As we approach the end of the discussion period, we kindly invite you to share any additional thoughts regarding our response to your concerns above.  We sincerely appreciate your efforts and valuable feedback thus far.
> >
> > We would also like to reiterate the core contribution of our paper.  This work is the first to identify a significant phenomenon: that all existing factual enhancement methods—whether training-based, inference-based, or prompt-based—lead to a substantial decrease in context-faithfulness.  We provide extensive experimental validation and insightful analysis to support this finding.  We believe that this discovery is highly valuable for the field of LLM research, offering key insights into the interplay between factuality and context fidelity.  Furthermore, it brings important considerations regarding knowledge conflicts and trade-offs in LLM alignment to the community.  Our proposed interpretative analysis lays a foundation for future work, ensuring that these findings will have both immediate and long-term impacts on advancing research in the field.
> >
> > Thank you again for your time and thoughtful consideration.
> >
> > Best regards,
> > The Authors

---

### Meta-Review · Area_Chair_rxuU · 2024-12-15

**Metareview:**

This paper examines the trade-off between enhancing factual accuracy and maintaining context-faithfulness in LLMs. Through experiments and analysis of model internals, the authors find that methods like representation editing and contrastive decoding improve factuality but significantly harm context-faithfulness. The study highlights the limitations of current approaches and calls for research to mitigate these adverse effects while improving factual accuracy.

In general, all the reviewers appreciated the paper and also engaged in the discussion. I also appreciate the authors for their detailed responses, which led to increased scores. I would like the authors to address the following points in the revised manuscript:

1. One of the reviewers raised serious concerns about the "factuality enhancement" and the broader score in the paper. The authors attempted to address this with elaborative comments, some of which might not be directly relevant to the given context. I suggest adding the responses appropriately in the paper without deviating from the main claims

2. The other reviewer suggested that practical suggestions or potential strategies for mitigating these trade-offs would strengthen the contribution and provide guidance for future work. I think this is really crucial given that the paper presents a critical aspect of the balance between factual accuracy and faithfulness.

3. Please address the typos and writing issues pointed out by the reviewers.

**Additional Comments On Reviewer Discussion:**

In general, the paper receives appreciation from all the authors. One reviewer was engaged in the discussion and made adjustments. The discussion mostly centred on the factuality enhancement, differences with the retrieval models, over-claims etc., most of which were addressed by the authors.

---

### Decision · Program_Chairs · 2025-01-22

Accept (Poster)